# PLK1 has tumor-suppressive potential in APC-truncated colon cancer cells

Monika Raab[1], Mourad Sanhaji[1], Yves Matthess[1,2], Albrecht Hörlin[3], Ioana Lorenz[1], Christina Dötsch[1], Nils Habbe[4], Oliver Waidmann[5], Elisabeth Kurunci-Csacsko[1], Ron Firestein[6,7], Sven Becker[1] & Klaus Strebhardt[1,2]

The spindle assembly checkpoint (SAC) acts as a molecular safeguard in ensuring faithful chromosome transmission during mitosis, which is regulated by a complex interplay between phosphatases and kinases including PLK1. Adenomatous polyposis coli (APC) germline mutations cause aneuploidy and are responsible for familial adenomatous polyposis (FAP). Here we study the role of PLK1 in colon cancer cells with chromosomal instability promoted by APC truncation (APC-ΔC). The expression of APC-ΔC in colon cells reduces the accumulation of mitotic cells upon PLK1 inhibition, accelerates mitotic exit and increases the survival of cells with enhanced chromosomal abnormalities. The inhibition of PLK1 in mitotic, APC-ΔC-expressing cells reduces the kinetochore levels of Aurora B and hampers the recruitment of SAC component suggesting a compromised mitotic checkpoint. Furthermore, *Plk1* inhibition (RNAi, pharmacological compounds) promotes the development of adenomatous polyps in two independent *Apc*^*Min/+* mouse models. High PLK1 expression increases the survival of colon cancer patients expressing a truncated APC significantly.

[1] Department of Gynecology, Goethe-University, 60590 Frankfurt, Germany. [2] German Cancer Consortium (DKTK)/ German Cancer Research Center, 69120 Heidelberg, Germany. [3] Institute of Pathology at the Department of Pathology, Goethe-University, 60590 Frankfurt, Germany. [4] Department of General and Visceral Surgery, Goethe-University, 60590 Frankfurt, Germany. [5] Department of Gastroenterology and Hepatology, Goethe-University, 60590 Frankfurt, Germany. [6] Centre for Cancer Research, Hudson Institute of Medical Research, Clayton, AU 31681, Australia. [7] Department of Molecular Translational Medicine, Monash University, Clayton, VIC 3800, Australia. These authors contributed equally: Monika Raab, Mourad Sanhaji. Correspondence and requests for materials should be addressed to K.S. (email: strebhardt@em.uni-frankfurt.de)

Genomic instability is a characteristic of almost all human cancers. Chromosomal instability (CIN) represents the most frequent form of genomic instability, which correlates to a high rate by which chromosome structure and number changes over time in cancer cells compared to normal cells.In hereditary types of cancer characterized by the presence of CIN, mutations in DNA repair genes have been correlated to genomic instability. In addition mutations in mitotic checkpoint genes in sporadic cancer are supporters of genomic instability. However, mutations in the mitotic checkpoint gene budding uninhibited benzimidazole 1 (BUB1) can induce CIN in cancer cell lines, but the frequency of Bub1 mutations in primary cancer tissues is low[1].

Colorectal cancer (CRC) is the second most frequent type of cancer with one million new cases diagnosed per year worldwide. Due to CIN ~85% of CRC are aneuploid[2]. Patients with a familial risk make up ~20% of all patients with CRC[3]. Hereditary cancer syndromes are divided into two categories based on the presence of polyposis, as exemplified by familial adenomatous polyposis (FAP) and hereditary nonpolyposis colorectal cancer (HNPCC). Germline mutations in the adenomatous polyposis coli (APC) gene are the cause for FAP. In sporadic colorectal cancer the APC gene is mutated in 80% of all cases, which harbor mutations in both alleles[4]. However, although both alleles are mutated in APC-defective human colorectal cancer cells, APC expression is not lost completely, typically N-terminal fragments of the APC protein are still being expressed[5].

The APC protein has the ability to bind a variety of proteins including microtubules, the cytoskeletal regulators EB1 and IQGAP1, components of the WNT/WG pathway β-Catenin and axin, and the RAC guanine-nucleotide-exchange factor (GEF) Asef1[6]. The majority of cancer-related APC mutations was detected in a region dubbed mutation cluster region (MCR) resulting in a carboxyterminal truncation[7]. The deleted region, that contains domains for the association with β-Catenin and microtubules, has been considered essential for the tumor suppressor activity of APC. APC has a well-established function as a negative regulator of the WNT/β-Catenin pathway by promoting degradation of β-Catenin[8]. Loss of APC is associated with the accumulation of β-Catenin in the nucleus, which activates the T-cell factor (TCF) and the lymphoid enhancer factor (LEF) transcription factor as targets of the canonical Wnt pathway[9,10]. Various lines of evidence support the model that a partial loss of APC function leads to the activation of the canonical WNT pathway, which is sufficient for intestinal tumorigenesis.

In humans, Polo-like kinase 1 (PLK1) controls multiple stages of cell-cycle progression. PLK1 is characterized by a C-terminal Polo-Box domain (PBD), which mediates protein interactions, the subcellular localization and regulates the N-terminal serine/threonine kinase domain[11,12]. PLK1 is responsible for a broad spectrum of cellular functions. It plays key roles for centrosome maturation[13], Golgi fragmentation[14], spindle assembly and function[15,16], kinetochore function[17,18], centromere assembly[19] and cytokinesis[20]. It also promotes DNA replication[21], mitotic entry[22], removal of sister chromatid cohesion[23], chromosome condensation[24] and APC/C activity[25]. PLK1 was found to be overexpressed in many types of human tumors[26,27]. In human colorectal cancer, PLK1 is expressed at higher levels in tumors compared to paired normal mucosa from the same patient in several independent studies[28,29], and the degree of overexpression correlates with adverse prognosis[30]. Remarkably, the analysis of PLK1-depletion in colon cancer cells in culture and in an inducible RNAi model in transgenic mice demonstrated that cancer cells and primary cells differ clearly in their dependency to PLK1 supporting a key role for PLK1 in colorectal carcinogenesis[15,31,32]. In our study on potential predictors of

radiation responsiveness, PLK1 expression was evaluated by immunohistochemistry ($n = 76$) or Affymetrix HG133 microarray ($n = 20$) on pretreatment biopsies of patients with advanced rectal cancer[33]. High PLK1 expression was associated with reduced overall patient survival ($p = 0.059$), and with less response to radiation, as graded in the resected specimen ($p = 0.004$), respectively.

Genetic instability, which includes both numerical and structural chromosomal abnormalities, is a hallmark of cancer. To explore the role of PLK1 in colon cancer, we hampered its function in a well-established, APC-related colon cancer model by using pharmacological means or RNAi. Here, we show that the mitotic arrest induced by PLK1 inhibition is weakened in APC-ΔC expressing colon cells accompanied by accelerated mitotic exit and improved cell survival. The inhibition of PLK1 in mitotic, APC-ΔC expressing cells compromised the mitotic checkpoint and accelerates the development of adenomatous polyps in two independent $Apc^{Min/+}$ mouse models. These finding support a 'tumor-suppressor function' for PLK1 in APC-ΔC expressing colon cells.

## Results

**Truncated APC can override PLK1-mediated mitotic arrest**. Based on the essential role of PLK1 during mitosis of all proliferating cells and its enriched expression in human cancer tissues, we set out for the investigation of the role of PLK1 in genetically unstable cancer. As a well-defined model system we used specific aneuploid colon cancer cells, because several studies have demonstrated that APC mutations resulting in the expression of N-terminal fragments cause CIN in human colon cancer[34–36]. Two categories of colorectal tumor cell lines were chosen: (1) the human epithelial colon cancer cell line HCT116 that was previously characterized to have a relatively stable genome (CIN⁻ cells, the HCT116 cell line has two wild-type APC alleles, is near-diploid, is chromosomally stable and has a robust spindle checkpoint[37]) and (2) the SW480 cell line that was previously characterized to have high rates of CIN (CIN⁺ cells; SW480, this cell line expresses a truncated APC version, N1338 and a wild-type APC protein[38,39]). To investigate the role of PLK1 in cancer cells with CIN, we expressed in both cell lines a truncated APC protein containing amino acids 1-750 of APC, hereafter named APC-ΔC, thus removing all β-Catenin regulatory sequences (Fig. 1a, b, upper left panels). We treated CIN⁻ HCT116, CIN⁺ SW480 cells and their APC-ΔC-expressing counterparts with the clinical PLK1 inhibitor BI6727[40] for 24–72 h and then analyzed the cell cycle status by western blot, by FACS and by determination of the mitotic indices. Upon BI6727 treatment the levels of Cyclin B1, PLK1, Aurora B and phospho-Histone H3 (pH3) accumulated up to 24 h in APC-ΔC-expressing and in control HCT116 cells, which is characteristic for a strong mitotic arrest induced by PLK1 inhibition (Fig. 1a, upper right panel). In APC-ΔC-expressing, BI6727-treated cells predominantly the levels of Cyclin B1 and PLK1 fell rapidly as seen at 48 h and 72 h compared to control cells (Fig. 1a, upper right panel). Similarly, the expression of APC-ΔC reduced the accumulation of HCT116 cells in the $G_2/M$ phases following exposure to BI6727 i.e. after an incubation period for 72 h the percentage of APC-ΔC-expressing HCT116 cells in $G_2/M$ (45%) differed significantly from Flag control cells (61%) ($P < 0.01$) (Fig. 1a, lower left panel). To corroborate our observations, we determined the mitotic index by assessing pH3-stained cells using FACS. The mitotic indices of HCT116 cells with and without APC-ΔC expression treated with 100 nM BI6727 peaked at 20–24 h. At 24–72 h the mitotic indices of APC-ΔC-expressing cells were below the levels of the corresponding control cells (24 h, 54% vs.

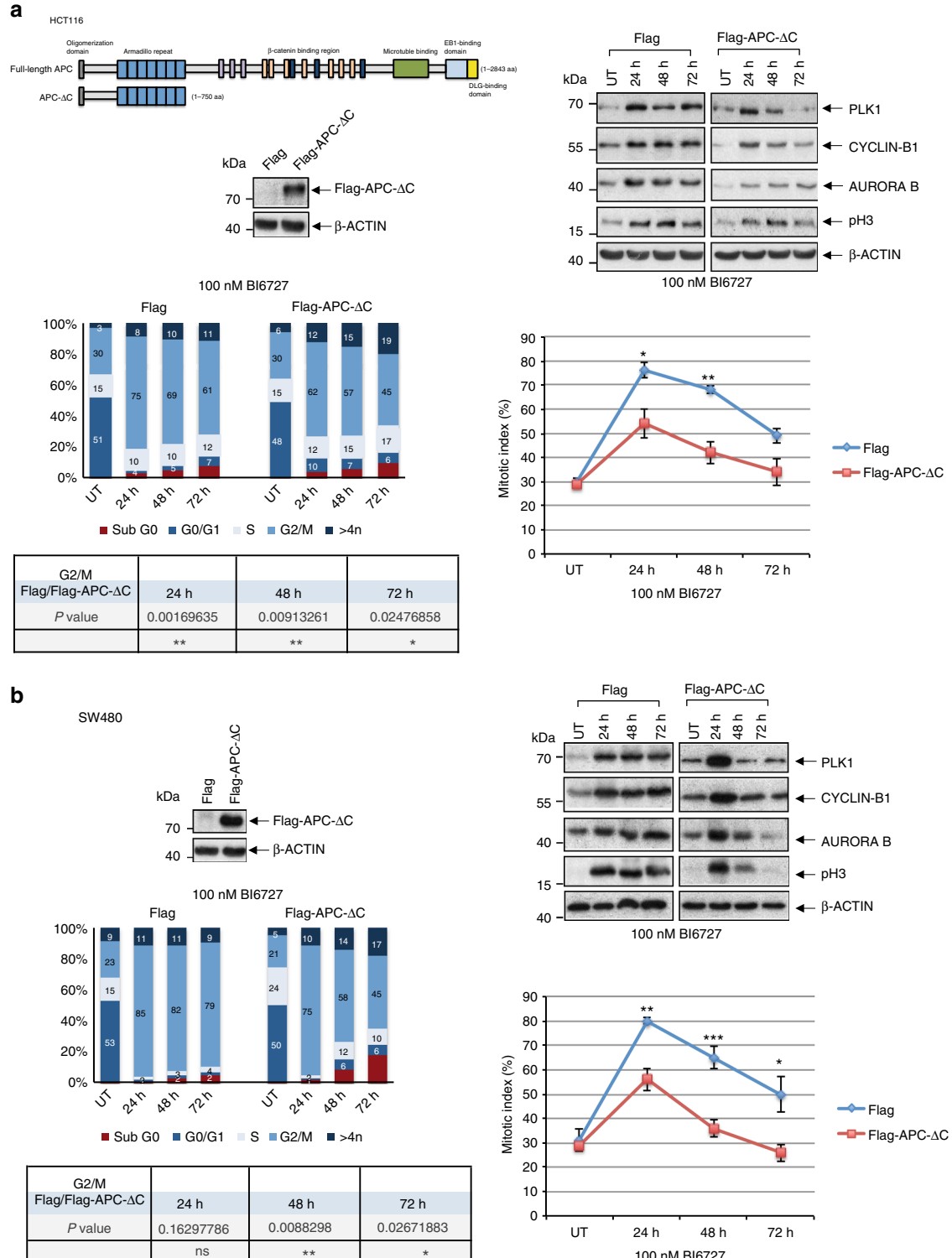

**Fig. 1** Correlation of PLK1 inhibition and mitotic arrest in APC-ΔC-expressing colon cells. (Upper left) Schematic representation of full-length APC and the truncated form APC-ΔC. Protein extracts from transfected colon cancer cell lines (**a**) HCT116 and (**b**) SW480 were blotted to detect Flag-tagged proteins. (Upper right) Cells were treated with 100 nM BI6727 followed by immunoblotting for PLK1, Cyclin B1, Aurora B, phosho-Histone H3 (pH3), and β-Actin. (Lower left) For the analysis of the cell cycle stage distribution of treated cells propidiumiodide was added to lysates which binds stoichiometrically to nucleic acids. After RNase-treatment and eliminating doublets or clumps the DNA-histogram determined by FACS gives information about the cell cycle. The representative quantification of the cell cycle analysis by FACS showing control (UT, untreated) and APC-ΔC-expressing cells treated for 24, 48, and 72 h with 100 nM BI6727; statistical evaluation of the percentages of cells in $G_2/M$ comparing APC-ΔC-expressing and control cells at 24, 48, and 72 h. (Lower right) The mitotic indices of cells treated with 100 nM BI6727 for 24, 48, and 72 h is depicted (means ± s.d., $n = 3$, for each concentration). *$P < 0.05$, **$P < 0.01$, ***$P < 0.001$, Student's $t$-test, unpaired and two-tailed

76%, $P < 0.05$; 48 h, 42% vs. 68%, $P < 0.01$; 72 h, 34% vs. 49%) (Fig. 1a, lower right panel). Next, we tested the ability of BI6727 to accumulate CIN+ SW480 cells with or without APC-ΔC-expression in mitosis (Fig. 1b, upper left panel). In line with our observations in HCT116 cells, after a peak at 24 h the expression of APC-ΔC in BI6727-treated SW480 cells induced a rapid decline of mitotic markers (Cyclin B1, PLK1, Aurora B, and pH3) (Fig. 1b, upper right panel), of the percentage of cells in $G_2/M$ (72 h, 79% vs. 45%, $P < 0.001$) (Fig. 1b, lower left panel) and of the mitotic index (24 h, 80% vs. 56%, $P < 0.01$; 48 h, 65% vs. 36%, $P < 0.001$; 72 h, 50% vs. 26%, $P < 0.05$) (Fig. 1b, lower right panel) compared to control cells.

To further confirm the results, we treated HCT116 and SW480 cells (Flag control, APC-ΔC-expressing) with increasing concentrations of BI6727 (25–2000 nM) for 24 h and performed Western Blot and FACS analyses. In control and in APC-ΔC-expressing cells the levels of Cyclin B1 peaked between 100 and 1000 nM (Supplementary Fig. 1a, b, left panels). Throughout the entire range of concentrations, the levels of Cyclin B1, PLK1, pH3 and the proportion of cells in $G_2/M$ were lower in APC-ΔC-expressing cells compared to controls (Supplementary Fig. 1a, b upper and lower panels). Taken together, in BI6727-treated colon cancer cells the expression of APC-ΔC can reduce the accumulation of cells in mitosis.

To investigate whether the expression of APC-ΔC influences the proliferative activity of colon cells, we determined the growth curve of both cell lines. Over an observation period of 12 days the proliferative activity of HCT116 and SW480 cells with APC-ΔC expression was higher compared to controls (Supplementary Fig. 2a). In the presence of BI6727 APC-ΔC-expressing SW480 cells proliferated faster compared to controls (Supplementary Fig. 2b). Notably, after 4–5 days BI6727 treatment, APC-ΔC-expressing HCT116 cells also started to proliferate faster (Supplementary Fig. 2b).

Since small molecule inhibitors of protein kinases might have off-target effects due to the high conservation of the ATP-binding pocket, we used RNAi for the silencing of the PLK1 expression. The expression of APC-ΔC reduced the accumulation of the mitotic marker pH3 in SW480 cells treated with PLK1 siRNA (Supplementary Fig. 2c, left panel). In addition, PLK1 depletion accelerated the growth of APC-ΔC-expressing SW480 cells compared to untreated controls (Supplementary Fig. 2c, right panel). Thus, experiments using a RNAi-based inhibition of PLK1 corroborated the data obtained using small-molecule PLK1 inhibitors.

Previous studies indicated that mitotic slippage occurs via slow Cyclin B1 degradation that is concomitant with defective mitotic spindle assembly checkpoint activity (SAC)[41]. Thus, the reduced ability of the PLK1 inhibitor BI6727 to arrest APC-ΔC-expressing colon cells in mitosis could be due to a compromised SAC.

**APC-ΔC expression accelerates exit from PLK1-induced arrest.** To investigate the mechanisms that contribute to the reduced mitotic index, as well as to assess the fate of mitotic cells in APC-ΔC-expressing cells treated with BI6727, we analyzed SAC activity by monitoring the mitotic exit. Towards this, HCT116 cells stably transfected with mCherry-H2B were synchronized in the S-phase, released in media with or without 100 nM BI6727 and followed by time-lapse microscopy (Fig. 2a). While control cells completed mitosis normally, APC-ΔC expression significantly abbreviated mitotic duration and showed an enhanced segregation failure during anaphase (Fig. 2b–d), which is in line with the role of APC in chromosome segregation[35]. The inhibition of PLK1 in both HCT116 and HCT116 APC-ΔC-expressing cells caused a prolonged mitotic arrest followed by mitotic escape

and the formation of polyploid cells. However, under BI6727, the expression of APC-ΔC significantly shortens the time in mitotic arrest compared to HCT116 cells (7.51 h vs. 10.9 h) and raised the endoreduplication in these cells (Fig. 2b–d). To shed more light on the cell fate induced by PLK1 inhibition we tracked the outcome of single cells during the mitotic arrest and after escaping mitosis. 22% of HCT116 died during mitotic arrest, 74% could evade mitosis but died in the next interphase and only 4% could escape mitosis and survive (Fig. 2b, e). In contrast to this, upon PLK1 inhibition, APC-ΔC-expressing cells showed a decrease in mitotic deaths down to 6%, and increased the fraction of cells that escaped mitosis and were able to survive, up to 46% (Fig. 2b, e). These results suggest that the expression of the truncated form APC-ΔC and the inhibition of PLK1 act synergistically to weaken the spindle assembly checkpoint. This, in turn, promotes the premature and improper mitotic exit, which generates cell death resistant polyploid cells. In addition to this aspect, two enrichment steps ((1) treatment with 100 nM BI6727 for 48 h, (2) mitotic shake-off) were performed to obtain a high percentage of mitotic cells. Subsequently, cells were replated for 8 h either in 100 nM BI6727 (non-released) or in PLK1 inhibitor-free medium (released) followed by monitoring the mitotic index (Supplementary Fig. 3). Under non-released conditions the mitotic index of Flag-transfected HCT116 control cells fell within 8 h to 83% compared to 71% in APC-ΔC-expressing cells (Supplementary Fig. 3a, left panel). Under released conditions the mitotic indexes differed between 50% (Flag-transfected) and 30% (APC-ΔC-expressing) (Supplementary Fig. 3a, left panel). In the western blot representing the kinetics of the mitotic marker proteins PLK1, Cyclin B1, and pH3, declined faster in APC-ΔC-expressing cells compared to control Flag-transfected cells (Supplementary Fig. 3a, right panel). In APC-ΔC-expressing SW480 cells (8 h after replating of shake-off cells) the mitotic index dropped under non-released conditions to 77% compared to 85% in control cells and under released conditions to 53% compared to 72% in control cells (Supplementary Fig. 3b, left panel). An accelerated decline of mitotic markers in SW480 cells expressing APC-ΔC was also observed (Supplementary Fig. 3b, right panel). SW480 cells exited mitosis more slowly than HCT116 cells (Supplementary Fig. 3a, b). This could be due to higher levels of Cyclin B1 in SW480 cells (Supplementary Fig. 3c), because CDK1 activity and the levels of Cyclin B1 are important parameters for timing of the mitotic exit. Taken together, the expression of APC-ΔC expedited the exit from mitosis in the presence or absence of the PLK1 inhibitor BI6727 indicating that APC-ΔC-expression compromises the SAC.

**APC-ΔC and polyploidy/survival of BI6727-treated cells.** To investigate whether an aberrant exit from mitosis due to a treatment with a targeted kinase inhibitor might cause SAC adaptation and subsequently aneuploidy, we tested the impact of PLK1 inhibitor treatment on the ploidy status of cells expressing APC-ΔC. Briefly, both cell lines with or without APC-ΔC-expression were treated with 100 nM BI6727 for 24–72 h and the DNA content was determined by FACS. While the expression of APC-ΔC in CIN HCT116 cells for 24 h led to an increase of the percentage of polyploid cells from 3 to 8%, the BI6727-treatment and APC-ΔC-expression of HCT116 cells induced an increase from 7 to 12% ($P < 0.05$; Fig. 3a). The same treatment in CIN+ SW480 cells induced an increase from 9 to 15% and from 11 to 20% in APC-ΔC-expressing SW480 cells, respectively ($P < 0.01$; Fig. 3a). To confirm the observations, we treated cells with 100 nM BI6727 for 24–72 h. While this treatment increased the percentage of polyploid HCT116 cells from 3% to 11% within 72 h, in APC-ΔC-expressing HCT116 cells an increase from 8 to 18%

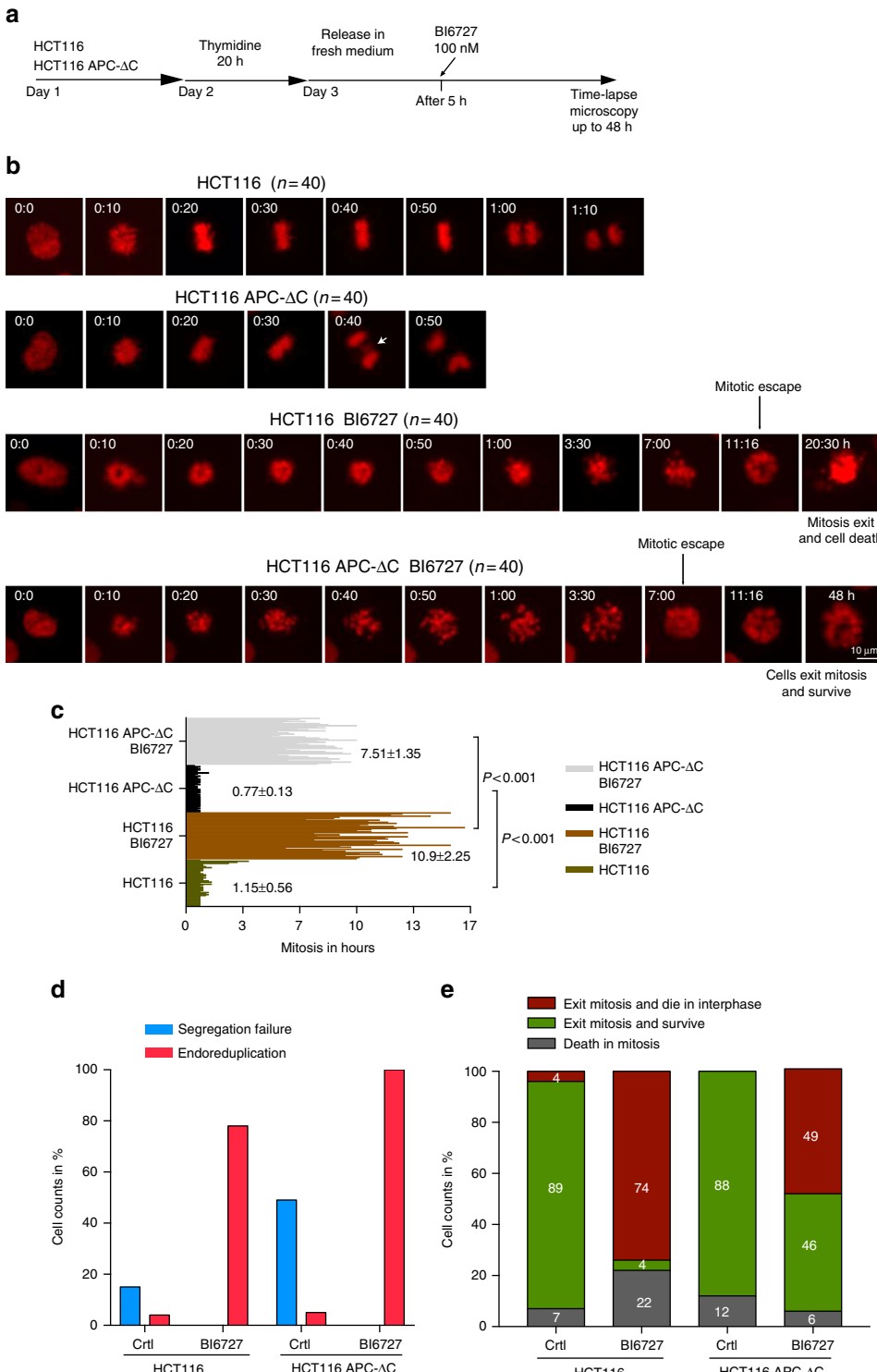

**Fig. 2** PLK1 inhibition shortens mitosis, increased polyploidy and enhances cell survival in APC-ΔC-expressing HCT116 cells. **a** Treatment schedule for time-lapse microscopy. Cells were first incubated with 2 mM thymidine to arrest them in the S-phase. After releasing the cells in fresh medium for 5 h, 100 nM BI6727 were added to cells and time-lapse microscopy was started up to 48 h after PLK1 inhibition. **b** Representative time-lapse of HCT116 and HCT116 expressing APC-ΔC mcherry-H2B treated as in (**a**) with or without 100 nM BI6727 (from $t = 0$ h to $t = 48$ h). The time of mitotic escape is indicated in the case of BI6727 treatment. The white arrow indicates lagging chromosomes observed during anaphase. Scale bar: 10 μm. **c** Duration of mitosis in HCT116 and HCT116 expressing APC-ΔC treated with or without BI6727 was assessed by time-lapse microscopy. Mitotic duration of each cell in the different treatment groups ($n = 40$) is quantitated in the bar graph. ***$P < 0.001$. **d** Segregation failure and endoreduplication rate observed in the different treatment groups in percentage. **e** Cell fate profile of HCT116 and HCT116 expressing APC-ΔC treated with or without 100 nM BI6727.

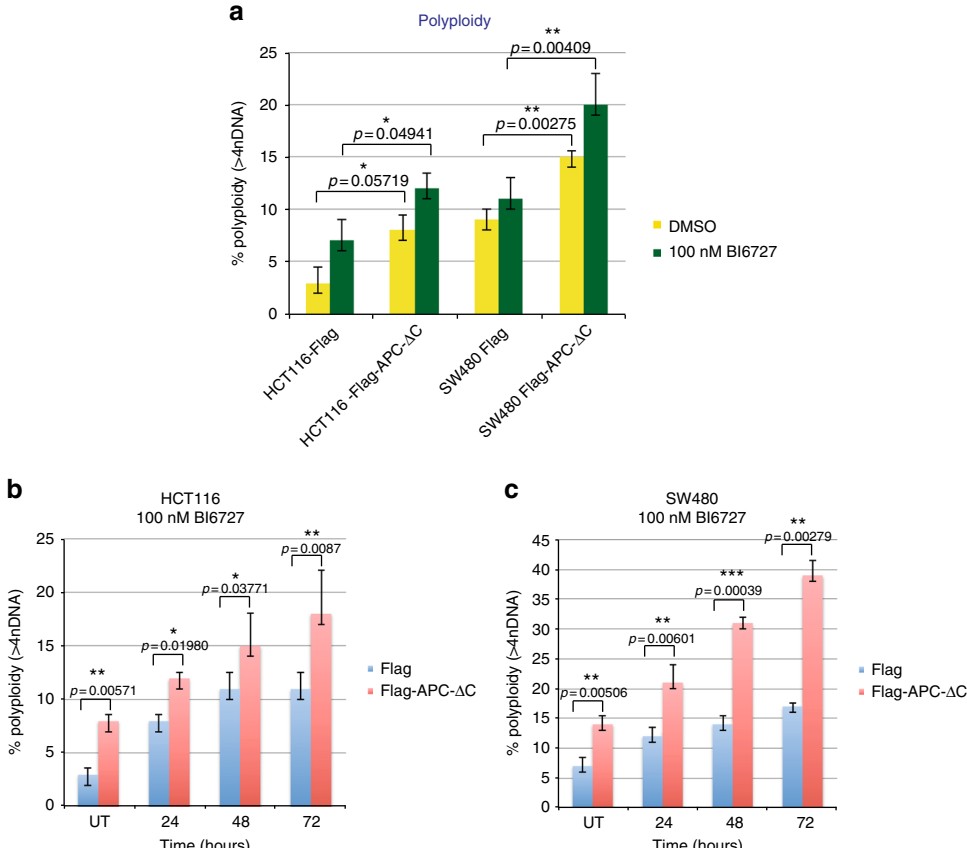

**Fig. 3** Chromosomal content of BI6727-treated, APC-ΔC-expressing colon cells. **a** 300,000 cells (HCT116, SW480) with or without APC-ΔC-expression treated for 24 h with 100 nM BI6727 were then subjected to DAPI staining and examined by FACS for the analysis of the DNA content. **b** HCT116 and (**c**) SW480 cells with or without APC-ΔC-expression were treated for 24, 48, and 72 h with 100 nM BI6727 and were subsequently subjected to DAPI staining and examination by FACS for the analysis of the DNA content (means ± s.d., $n = 3$, for each time point). $*P < 0.05$, $**P < 0.01$, $***P < 0.001$, Student's $t$-test, unpaired and two-tailed

could be measured (Fig. 3b). Under the same treatment conditions the percentage of polyploid SW480 cells rose from 7 to 17% within 72 h, in APC-ΔC-expressing SW480 cells an increase from 14 to 39% was detected (Fig. 3c). Taken together, while the incubation of both cell lines with BI6727 increased the percentage of polyploid cells significantly, the simultaneous expression of APC-ΔC increased the endoreduplication in cells even further.

To study the tumorigenic potential of cells exhibiting endoreduplication, we tested the ability of cells to survive and to form colonies. Briefly, for this colony formation assay. Cells with or without APC-ΔC expression were cultured for 2 days in the presence of BI6727 and then 1200 cells were re-plated in 6 cm-dishes in inhibitor-free medium (Fig. 4a). After 7 days the resulting colonies were stained with Coomassie blue and counted (Fig. 4b, c, left and middle panels). Remarkably, the expression of APC-ΔC increased the number of BI6727-treated cells significantly ($P < 0.001$) (Fig. 4b, c, middle panels). The corresponding investigation of apoptotic activity in HCT116 and SW480 cells treated with 100 nM BI6727 revealed at 48–72 h and in a range of concentrations between 25 and 2000 nM BI6727 more caspase 3/7 activity and more annexin staining compared to their APC-ΔC-expressing counterparts (Supplementary Fig. 4a, b) providing evidence for a higher rate of apoptosis in control cells compared to APC-ΔC-expressing cells. In APC-ΔC-expressing, BI6727-treated cells (HCT116, SW480) the growth rate showed a strong increase in relative cell numbers compared to control cells reaching maximal levels at day 5–7 (Fig. 4b, c, right panels).

To elucidate, whether PLK1 inhibition modulates the APC/β-Catenin pathway, we first expressed APC-ΔC in HCT116 and SW480 cells. This led to an increase in the level of β-Catenin predominantely in HCT cells (Fig. 4d, left panel). In both cell lines the downstram targets of β-Catenin, MYC and Cyclin D, were increased. Moreover, we inhibited PLK1 function by using BI6727 or PLK1-specific siRNA for 24 h (Fig. 4d, right panel). This inhibition did not alter the level of β-Catenin confirming previous data[42]. Taken together, the results of the long-term experiments indicate reduced apoptosis associated with a higher colony forming ability of APC-ΔC-expressors in the presence of BI6727 compared to control cells. However, PLK1 does not seem to regulate the APC/β-Catenin pathway by modulating the stability of β-catenin.

**PLK1 inhibition in APC-ΔC cells increases chromosomal anomaly.** The ability to assess chromosome abnormalities based on the analysis of metaphase chromosome spreads can reveal genetic disorders in cells with or without APC-ΔC expression that have survived the treatment with BI6727. To this end, near diploid HCT116 and SW480 cells that have an average number of chromosomes of 56 (range, 52–58)[43] were treated for 2 days with BI6727, thereafter transferred to compound-free medium followed by examination of metaphase spreads of the surviving cells (Fig. 5a, Supplementary Fig. 5a). The mean number of chromosomes per cell in the HCT116 population increased from $42.2 \pm 3.35$ to $63.24 \pm 5.59$ in cells that survived

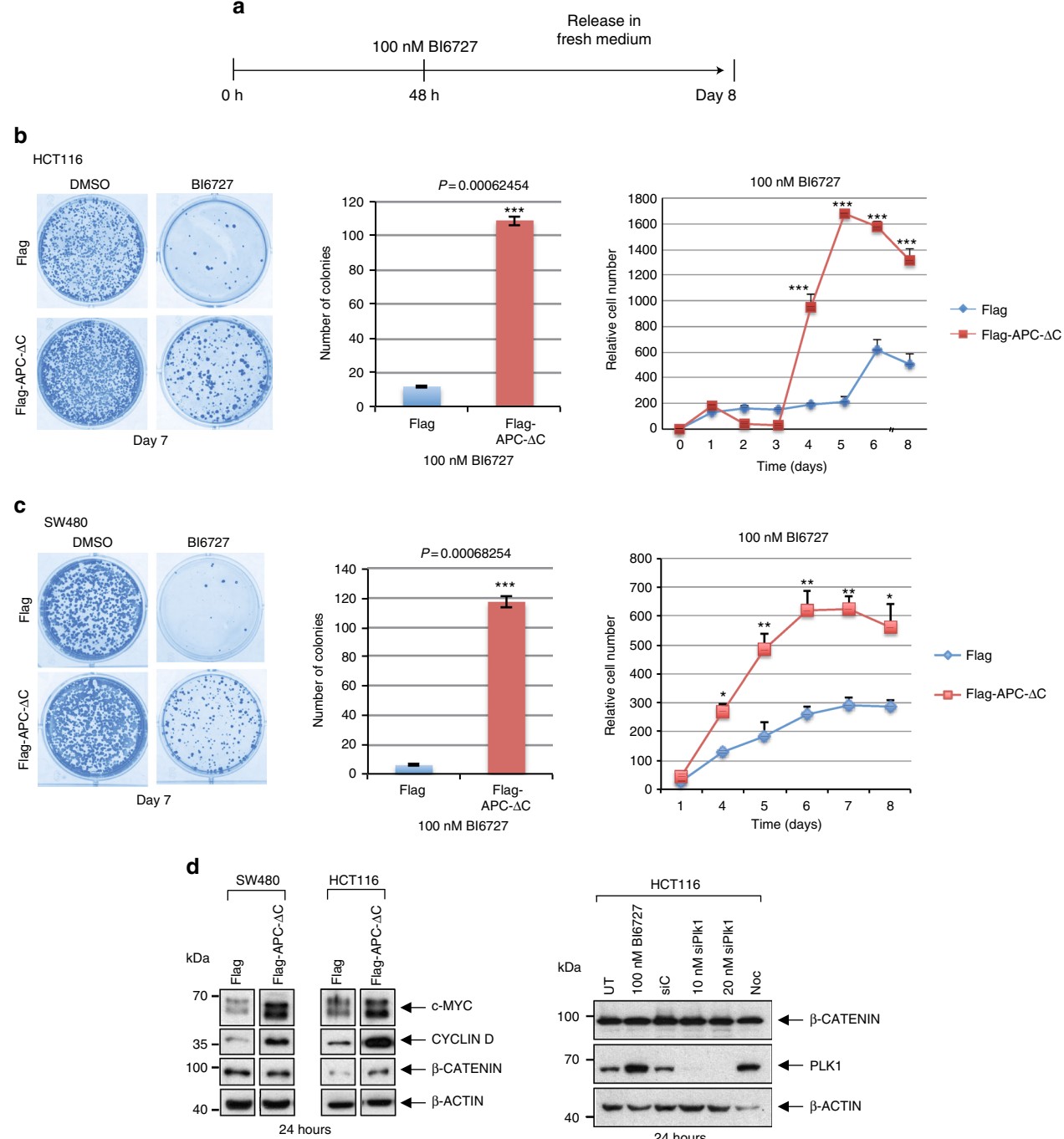

**Fig. 4** Survival analysis of BI6727-treated, APC-ΔC-expressing colon cells. **a** Scheme of the experimental procedure. HCT116 and SW480 cells with or without APC-ΔC-expression were incubated in the presence of BI6727 (100 nM) for 48 h. On day 2, BI6727 was washed away, 1200 cells were replated in fresh medium and cultivated for 8 days. **b** HCT116 and (**c**) SW480 (Left) cells were subjected to Coomassie Blue staining and (Middle) the number of colonies was determined. (Right) The proliferative activity was assayed using an MTT assay, which is a method to quantify cell proliferation and viability. The assay is based on the conversion of water soluble MTT (3-(4,5-dimethylthiazol-2-yl)-2,5-diphenyltetrazolium bromide) compound to an insoluble formazan product. Viable cells with active metabolism convert MTT into formazan. Dead cells, on the other hand, lose this ability and therefore show no signal. Thus, color formation serves as a marker of only the viable cells. Over a period of 8 days the numer of viable cells was quantified. (means ± s.d., $n =$ 3, for each time point). *$P < 0.05$, **$P < 0.01$, ***$P < 0.001$, Student's $t$-test, unpaired and two-tailed. **d** (Left) HCT116 and SW480 cells with or without APC-ΔC-expression were immunoblotted for c-MYC, Cyclin D, β-Catenin, and β-Actin. (Right) HCT116 cells were either untreated, treated with 100 nM BI6727, control siRNA siC, PLK1-specific siRNA siPLK1 or Noc and then immunoblotted for β-Catenin, PLK1, and β-Actin

the 48 h-treatment with 100 nM BI6727 (Fig. 5b, c). In APC-ΔC-expressing cells kept under the same conditions an increase from 47.1 ± 3.06 to 87.84 ± 9.9 was determined. While the same treatment (48 h, 100 nM BI6727) of CIN⁺ SW480 cells increased the

mean number of chromosomes from 50.54 ± 3.73 to 61.09 ± 2.89 at day 12 following treatment, in APC-ΔC-expressing cells a rise from 61.23 ± 5.55 to 84.95 ± 7.24 was determined (Supplementary Fig. 5b, c) suggesting that both treatments (1) expression of APC-

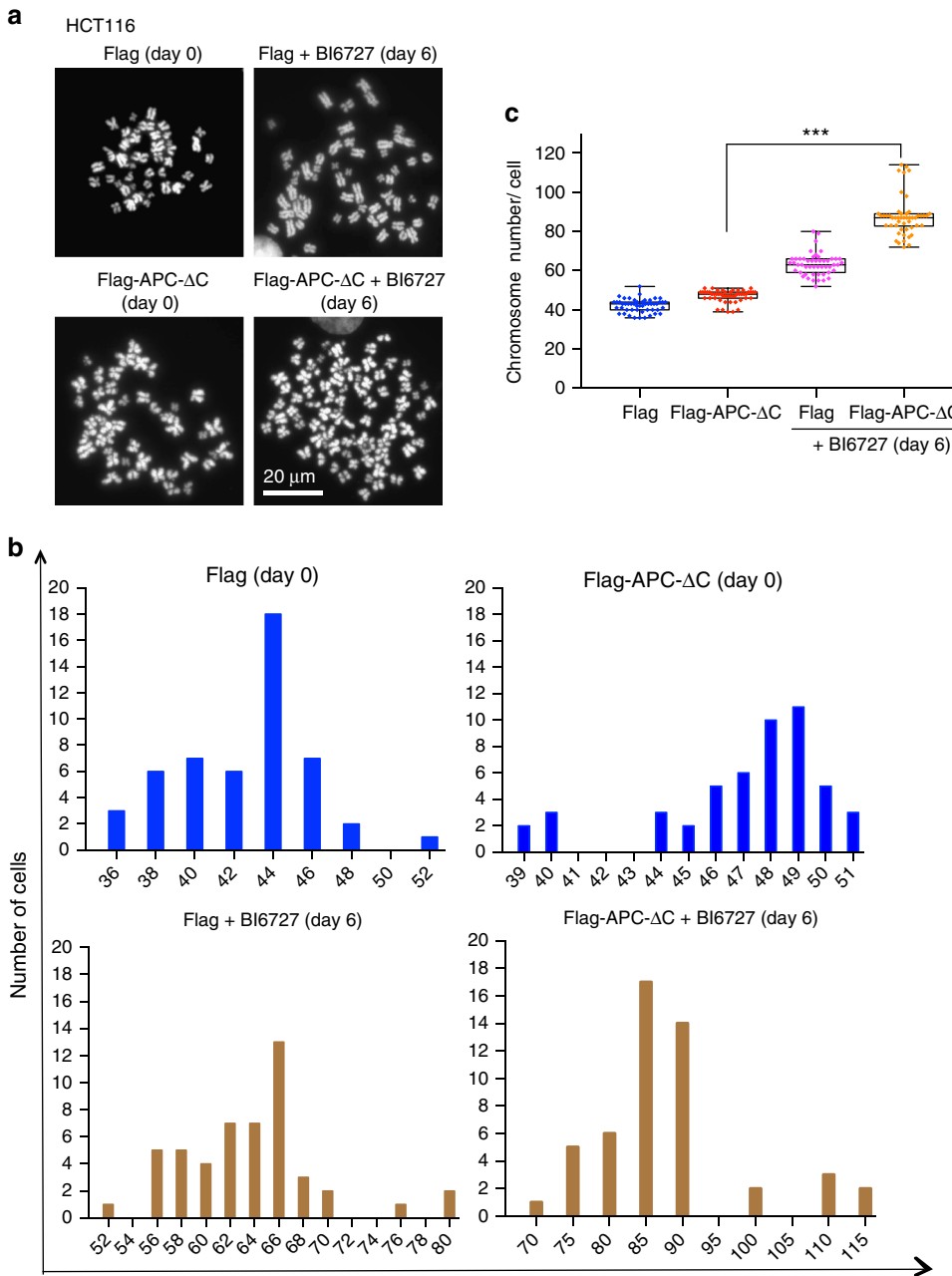

**Fig. 5** Analysis of chromosomal aberrations in BI6727-treated, APC-ΔC-expressing colon cells. **a** Following the incubation with BI6727 for 48 h and subsequent release into fresh medium for 6 days, metaphase spreads of HCT116 cells with or without APC-ΔC-expression were prepared and the chromosomes were stained with Hoechst. Scale bar: 20 μm. **b** The histogram plotting of the distribution of chromosome numbers at day 0 and day 6 is shown. **c** Quantification of the number of chromosomes in 500 HCT116 cells with or without APC-ΔC-expression incubated in the presence or absence of BI6727 (100 nM) for 48 h. (means ± s.d., $n = 3$, for each treatment). ***$P < 0.001$, Student's $t$-test, unpaired and two-tailed

ΔC and (2) PLK1 inhibition contribute to promote chromosomal abnormalities in colon cancer cells with different degrees of chromosomal instability.

**Inhibiting PLK1 in ΔAPC cells reduces Aurora B at kinetochores**. While APC-ΔC-expressing colon cells showed a wide range of karyotypes, the karyotypic complexity is reinforced by BI6727 treatment (Fig. 5, Supplementary Fig. 5) providing additional evidence for the existence of a compromised SAC. Considering the role of PLK1 and Aurora B in sustaining a functional checkpoint signaling during mitosis[44,45], we were interested to

investigate the localization and the activity of Aurora B in APC-ΔC expressing cells with or without PLK1 inhibition. Towards this end, we incubated APC-ΔC-expressing and control HCT116 cells with a reversible Eg5 inhibitor (STLC)[46] that induces a prometaphase arrest without interfering with microtubule dynamics, and with 100 nM BI6727 for 16 h followed by fixation, staining with Aurora B antibodies and quantification (Fig. 6a). The expression of APC-ΔC in HCT116 cells with unperturbed PLK1 activity induced a partial displacement of Aurora B from kinetochores compared to control cells (Fig. 6b, panels a–d vs. i–l).

The inhibition of PLK1 by BI6727 reduced also Aurora B kinetochore staining in both WT and APC-ΔC-expressing cells

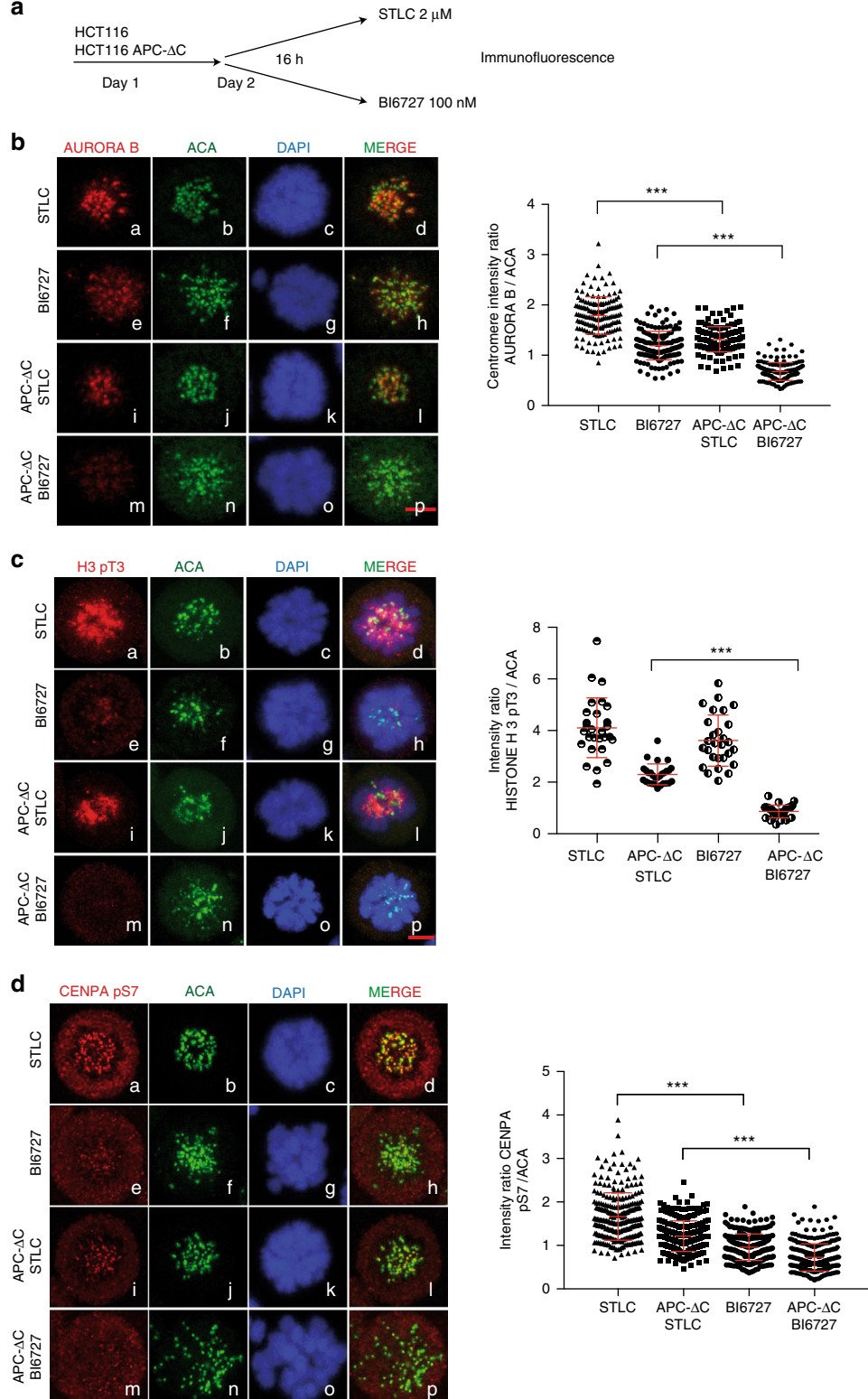

**Fig. 6** Inhibition of PLK1 decreases the levels of Aurora B, CENP-A pS7 and Histone 3 pT3 in cells expressing APC-ΔC. **a** Scheme of the experimental procedure. APC-ΔC-expressing and control HCT116 cells were seeded on day 1, treated on day 2 either with 2 μM STLC or 100 nM BI6727 for 16 h and harvested on day 3. Cells were fixed, processed for immunofluorescence using antibodies for (**b**) Aurora B, (**c**) Histone 3 pT3 (H3 pT3), (**d**) CENPA pS7and ACA. Intensities were normalized to ACA and quantified. Scale bar, 5 μm. Values were calculated from at least 50 cells per treatment. (means ± s.d., $n = 3$, for each treatment). ***$P < 0.001$, Mann-Whitney $U$-test

(Fig. 6b, panels e–h vs. m–p). Remarkably, Aurora B delocalization was most prominent in BI6727-treated, APC-ΔC-expressing cells, where Aurora B recruitment to kinetochores was almost completely abolished and only a diffuse staining of Aurora B was still visible (Fig. 6b, panels m–p). In order to understand whether the effect of APC-ΔC expression on Aurora B mislocalization is a consequence of errors in the assembly of kinetochore protein, we stained HCT116 cells arrested in mitosis with the kinetochore components ZW10 interacting protein 1 (ZWINT-1), HEC1 and KNL1, kinetochore-associated protein essential for accurate chromosome segregation in eukaryotic cells (Supplementary Fig. 6). Since the recruitment of HEC1 and KNL1 is dependent on kinetochore core components such as CENP-I[47], monitoring their recruitment provides an indication about the structural integrity of kinetochores. Interestingly, the kinetochore intensities of all investigated proteins was reduced upon expression of APC-ΔC compared to control cells (Supplementary Fig. 6b, c, d). This suggest that the sole expression of APC-ΔC alters the core structure of kinetochores. This recruitment was even more disturbed when APC-ΔC expression was combined to PLK1 inhibition. This suggests that the loss of PLK1 activity added to APC-ΔC acts synergistically towards preventing a normal kinetochore assembly.

The study of APC-ΔC-expressing and non-expressing SW480 cells confirmed the observation of a strong displacement of Aurora B from kinetochores in BI6727-treated, APC-ΔC-expressing cells (Supplementary Fig. 7b panels a–d vs. m–p).

The activity of the protein kinase Haspin plays an important role in recruiting the chromosome passenger complex (CPC) to kinetochores through phosphorylation of Histone H3 at Thr3 (H3pT3)[48]. Haspin is in turn activated by PLK1 phosphorylation[49]. Interfering with the PLK1-specific phosphorylation of Haspin using BI6727 might prevent, at least in part, the recruitment of Aurora B to kinetochores. To investigate this aspect, we fixed and stained APC-ΔC-expressing cells with antibodies against H3pT3. Whereas PLK1 inhibition reduced the intensity of H3pT3 at kinetochores in control HCT116 and SW480 cells (Fig. 6c, Supplementary Fig. 7c, panels a–d vs. e–h), APC-ΔC expression induced an even stronger reduction of the H3pT3 signal (Fig. 6c, Supplementary Fig. 7c, panels a–d vs. i–l). In BI6727-treated, APC-ΔC-expressing cells the H3 pT3 signal is almost completely gone (Fig. 6c, Supplementary Fig. 7c, panels m–p)

Furthermore, CENP-A pS7, an established marker for Aurora B activity, showed that this signal is reduced in APC-ΔC-expressing HCT116 and SW480 cells with unperturbed PLK1 activity compared to WT cells (Fig. 6d, a–d vs. i–l, Supplementary Fig. 7d, panels a–d vs. i–l). Remarkably, upon BI6727 treatment CENP-A pS7 becomes significantly weaker in APC-ΔC-expressing cells compared to controls (Fig. 6d, i–l vs. m–p, Supplementary Fig. 7d, panels i–l vs. m–p). The reduced activity of Aurora B in APC-ΔC-expressing cells under BI6727 treatment could be confirmed by an in vitro kinase assay using immunoprecipitated Aurora B with GST-MCAK as substrate and in a Western Blot analysis using CENP-A pS7 as a marker (Supplementary Fig. 7e, f). Taken together, these observations indicate that the expression of APC-ΔC decreases the levels of Aurora B at kinetochores. The inhibition of PLK1 in APC-ΔC-expressing cells reinforces this effect and leads to an almost complete loss of the kinetochore-bound Aurora B and consequently to the loss of its activity at these sites.

To investigate whether a compromised SAC induced by PLK1 inhibition is comparable to effects in APC-ΔC-expressing cells induced by the inhibition of Aurora B, we treated APC-ΔC-expressing cells with the small molecule Aurora B inhibitor AZD1152 and tested for the cellular response (Supplementary Fig. 8). We could confirm that the proliferative activity of

cells expressing APC-ΔC is higher compared to WT cells (Supplementary Fig. 8a, b). The treatment of APC-ΔC-expressing HCT116 cells with AZD1152 increased the proliferation further compared to untreated APC-ΔC-expressing controls (Supplementary Fig. 8b). Moreover, a prominent increase in the percentage of cells with a DNA content >4N and in the percentage of polyploid cells could be observed (Supplementary Fig. 8c, d). The treatment also increased the number of colonies of APC-ΔC-expressing cells (Supplementary Fig. 8e) suggesting that the inhibition of both mitotic regulators (PLK1, Aurora B) contribute to weaken SAC signaling in APC-ΔC-expressing cells.

**PLK1 inhibition impairs SAC activity in ΔAPC-expressing cells**. Our experiments revealed that the inhibition of PLK1 in APC-ΔC-expressing cells decreases the mitotic index (Figs. 1, 2) and reduced significantly the kinetochore level and activity of Aurora B (Fig. 6b, Supplementary Fig. 7b). This was accompanied by a strong increase in polyploidy in APC-ΔC-expressing cells (Figs. 3, 5, Supplementary Fig. 5). A possible cause for the elevated polyploidy encountered after PLK1 inhibition is mitotic slippage, which is a consequence of a defective SAC. In order to investigate the SAC functionality, we assessed the kinetochore localization of BUBR1 and MAD1, two components of the mitotic checkpoint signaling. Cells were treated with STLC or BI6727 for 16 h (Fig. 7a, Supplementary Fig. 9a). We observed that the expression of APC-ΔC in HCT116 and SW480 alone was sufficient to reduce the levels of BUBR1 (Fig. 7b, panels a-d vs. i-l, Supplementary Fig. 9b, panels a-d vs.i-l) and MAD1 (Fig. 7c, panels a-d vs.i-l, Supplementary Fig. 9b, panels a-d vs.i-l) at kinetochores compared to controls confirming previous findings[50]. Inhibition of PLK1 substantially reduced BUBR1 (Fig. 7b, panels a-d vs. e-h, Supplementary Fig. 9b, panels a-d vs. e-h) and MAD1 recruitment (Fig. 7c, panels a-d vs. e-h, Supplementary Fig. 9b, panels a-d vs. e-h) to kinetochores of APC-ΔC-expressing and control cells (Fig. 7c, panels a-d to m-p, Supplementary Fig. 9b, panels a-d to m-p). Remarkably, BUBR1 and MAD1 almost completely failed to localize to kinetochores of cells expressing APC-ΔC upon BI6727 treatment (Fig. 7b, panels i-l vs. m-p, Fig. 7c, panels i-l vs. m-p, Supplementary Fig. 9b, panels i-l vs. m-p) suggesting a compromised SAC. Additionally, we analyzed the mitotic checkpoint complex (MCC), which is composed of BUB3 together with MAD2 and BUBR1 bound to CDC20. We precipitated CDC20 using specific antibodies and investigated the co-precipitated protein. PLK1 inhibition in APC-ΔC-expressing cells caused a dramatic disassembly of the MCC suggesting a checkpoint override in these cells (Fig. 7d, Supplementary Fig. 9c). Thus, inhibiting PLK1 in APC-ΔC cells impairs the function of the mitotic checkpoint allowing the cells to improperly exit mitosis, hereby, generating aneuploid cells.

**PLK1 inhibition increases intestinal tumors in $Apc^{Min/+}$ mice**. This study has revealed that the inhibition of PLK1 in ΔAPC-expressing cells increases the number of surviving aneuploid cells (Fig. 4). To determine whether $Plk1$ is directly involved in intestinal carcinogenesis in living animals, we tested two independent mouse models: (1) $Apc^{Min/+}$ mice were treated with BI6727 and (2) to avoid selectivity issues of BI6727 caused by the conserved nature of the ATP-binding site in protein kinases, we used our RNA interference (RNAi)-based, murine model for an inducible knockdown (iKD) of $Plk1$[32] to generate $Apc^{Min/+}$ $Plk1^{iKD}$ mice. Briefly, the in vitro fertilization (IVF) technique was applied to obtain a population of $Apc^{Min/+}$ $Plk1^{iKD}$ mice with the desired genotype and a nearly identical age (Fig. 8a). C57BL/6J-ApcMin /J oocytes were fertilized with cryopreserved sperm

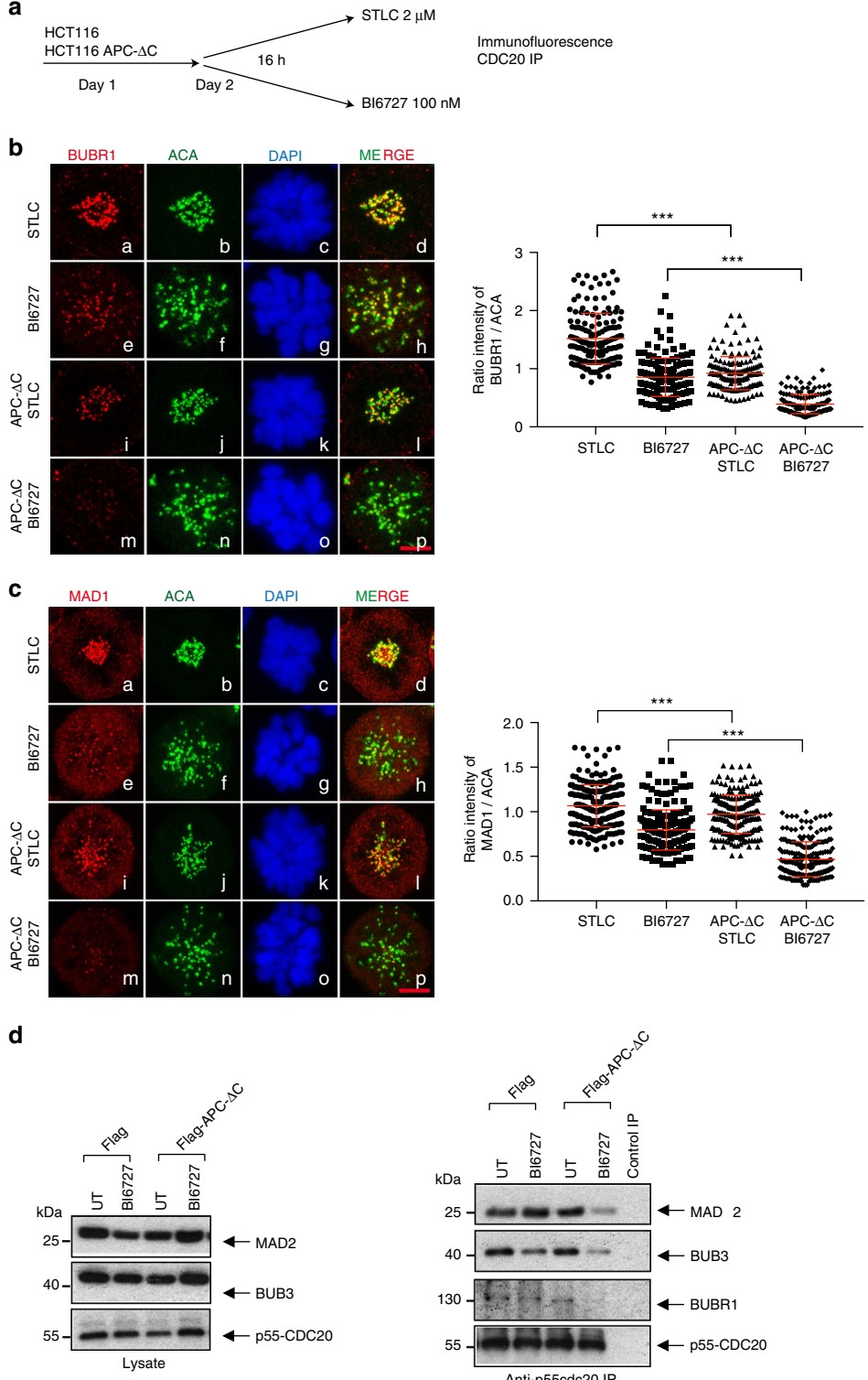

**Fig. 7** Inhibition of PLK1 in APC-ΔC-expressing HCT116 cells reduces the recruitment of BUBR1, MAD1 to kinetochore and decreases the mitotic checkpoint complex association. **a** Scheme of the experimental procedure. APC-ΔC-expressing and control HCT116 cells were seeded on day 1, treated on day 2 either with 2 μM STLC or 100 nM BI6727 for 16 h and harvested on day 3. The kinetochore recruitment of the checkpoint proteins (**b**) BUBR1 and (**c**) MAD1 was monitored. After treatment, cells were fixed and stained with the indicated antibodies. Scale bar, 5 μm. The kinetochore intensities of BUBR1 and MAD1 staining in the different treatment groups were quantified. The intensities were normalized to ACA. Values were calculated from at least 50 cells per treatment (means ± s.d., n = 3, for each treatment). ***P < 0.001, Student's t-test, unpaired and two-tailed. **d** Lysates (left panel) and immunoprecipitation of CCD20 from asynchronous and BI6727 (100 nM)-treated HCT-116 and APC-ΔC-expressing HCT116 cells were analyzed (right panel). CCD20-interacting proteins from IP were analyzed using western blot for MAD2, BUB3, BUBR1, and p55-CDC20

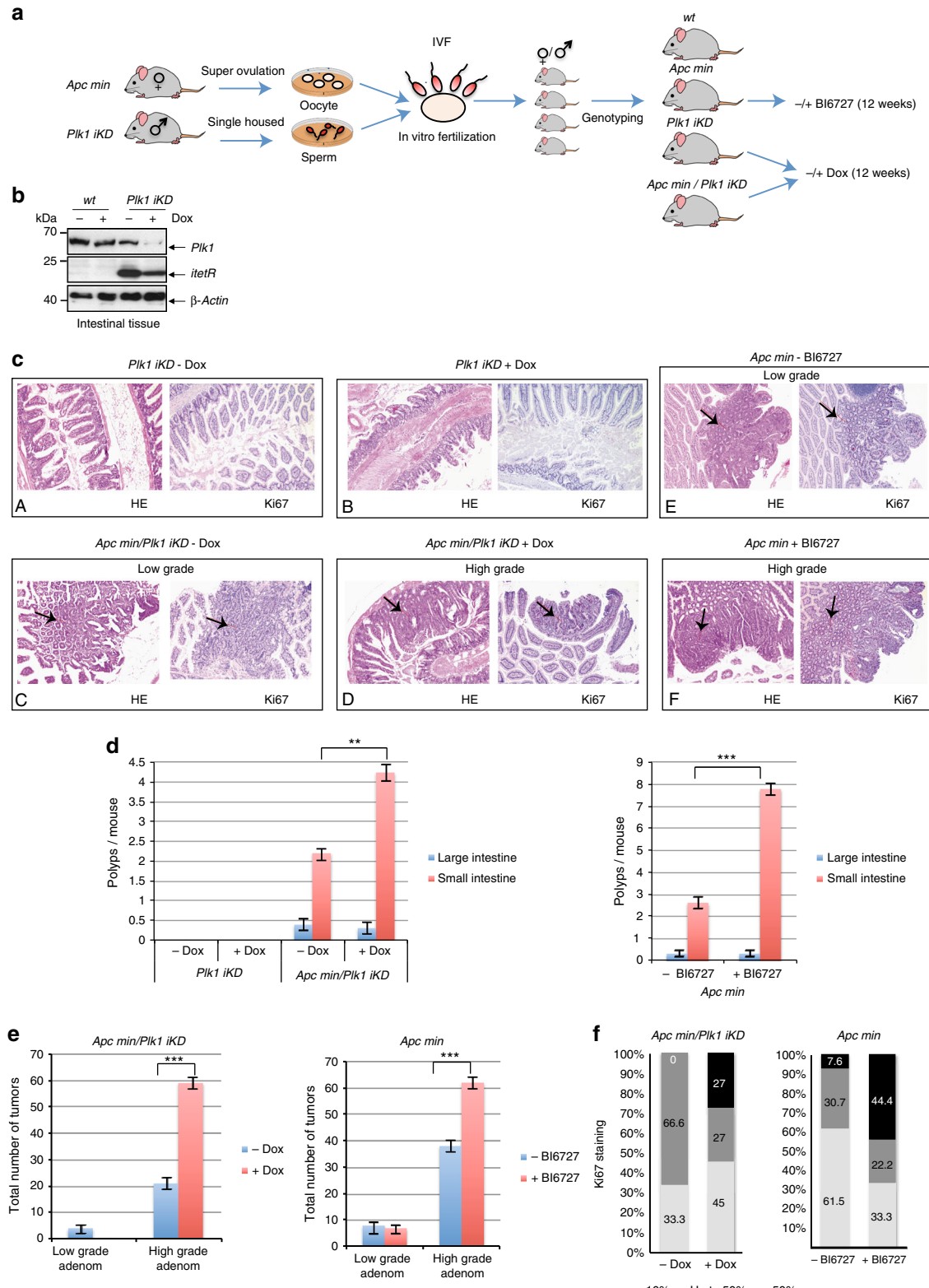

from C57BL/6;129S6-Gt(ROSA)26Sortm1226(RNAi:PLK1) ArteN3 *Plk1*[iKD] males. A cohort of 153 offspring was analyzed by genotyping. The frequency of newborn mice heterozygous for both *Apc*[Min/+] and *Plk1*[iKD] was 21.0% (32/153), which is close to the percentage predicated by Mendelian's law of segregation. Next, to determine whether primary mouse cells recapitulate the differential effect of Plk1 inhibition on cellular proliferation of

cells expressing truncated Apc, we examined the effects of gradual *Plk1* depletion or *Plk1* inhibition by BI6727 under controlled culture conditions. First, we identified mouse embryonic fibroblasts (MEF) clones representing different genotypes (*Apc*[Min/+], *Plk1*[iKD], *Apc*[Min/+] *Plk1*[iKD], wt) by PCR and Western blotting (Supplementary Fig. 10a). The Dox concentration (0–100 µg ml −1) and the *Plk1* protein levels were inversely correlated

(Supplementary Fig. 10b). Although *Plk1* depletion with 20–100 μg ml−1 Dox for 96 h had no significant impact on the proliferative activity of different iKD MEF clones, $Apc^{Min/+}$ $Plk1^{iKD}$ MEFs showed enhanced proliferation (Supplementary Fig. 10c, upper panels). Under the treatment of MEFs with BI6727 (10 nM, 100 nM) $Apc^{Min/+}$ $Plk1^{iKD}$ MEFs showed again an accelerated proliferation compared to wt MEFs (Supplementary Fig. 10c, lower panels).

At the age of 6–8 weeks we started to inhibit *Plk1* by RNAi or by BI6727-treatment in 3 genetically different cohorts ($Apc^{Min/+}$, $Plk1^{iKD}$, $Apc^{Min/+}$ $Plk1^{iKD}$) and monitored the impact on different relevant parameters including tumor development. In both experiments (BI6727 treatment, RNAi-based Plk1 silencing) *Plk1* inhibition did not show any considerable change in food consumption and gain in body weight compared with controls.

Following the termination of the experiment after 12 weeks of Dox-treatment, we used the Swiss roll technique[51] for the histological examination of the murine intestines. The analysis of the *Plk1* protein expression confirmed an efficient silencing in intestinal tissues with residual levels <10–20% (Fig. 8b). The histological examination revealed that $Plk1^{iKD}$ (±Dox) mice did not develop tumors, nor tubular adenomas neither high-grade tubular adenomas or adenocarcinomas (Fig. 8c, A, B). Because $Apc^{Min/+}$ mice typically develop adenomas in the small intestine by 3 months of age, we wondered how *Plk1* silencing influences tumor development. In $Apc^{Min/+}$ without BI6727- and $Apc^{Min/+}$ $Plk1^{iKD}$ mice without Dox-treatment a few well-defined tubular adenomas and numerous tubular adenomas were seen mostly in the small intestine and only a few in the colon. The well-defined adenomas (low grade adenoma) showed regular tubular proliferations with only slight atypia of cells and nuclei (Fig. 8c, C, E, black arrow). The high-grade adenomas (intramucosal adenocarcinoma) however showed irregular proliferated tubular villi and crypts with conspicuous atypia of the epithelia and nuclei and invasive growth within the lamina propria of mucosa (Fig. 8c, D, F, black arrow). No destructive growth beyond the mucosal muscularis was observed. The quantification revealed that Dox-treatment of age-matched $Apc^{Min/+}$ $Plk1^{iKD}$ mice did not change the number of polyps in the large intestine, but increased significantly the number of adenomas in small intestines (4.23 ± 0.3) compared to mice without Dox-treatment (2.2 ± 0.2) having normal levels of *Plk1* (Fig. 8d, left panel). In the parallel experiment using for the inhibition of *Plk1* the oral application of BI6727 in $Apc^{Min/+}$ mice, the treatment with BI6727 did not change the low number of adenomas/mouse in the large intestine (0.33 ± 0.1), but a significant increase was observed for the number of adenomas in the small intestine from 2.63 ± 0.2 per control mouse to 7.77 ± 0.3 in BI6727-treated animals representing an increase of 295% (Fig. 8d, right panel). Immunohistochemically, the tumor cells exhibited a strong staining for Ki67 in the nuclei within the tumor mass, whereas the adjacent normal mucosa showed nuclear positivity only in the epithelia of basal crypts (Fig. 8c). The histological analysis revealed that adenomas from Dox-treated $Apc^{Min/+}$ $Plk1^{iKD}$ mice and $Apc^{Min/+}$ mice treated with BI6727 progressed to much higher grades than those from untreated mice (Fig. 8e, left and right panels). Using proliferation marker Ki67-staining with cut-off values of <10%, ≤50% and >50% respectively, we observed a higher proliferative activity in tumors with *Plk1* inhibition (Fig. 8f).

**Colon cancer patients with low PLK1 have bad prognosis.** Based on our results in animal models, we hypothesized that PLK1 activity may impinge on the clinical outcome of colon cancer patients. To test this possibility, we utilized The Cancer Genome Atlas (TCGA) to identify 104 colon cancer patient samples harboring APC non-sense mutations for which PLK1 expression and clinical meta-data was also available. We segregated patients into PLK1-low and PLK1-high based on a median cutoff (Fig. 9, Supplementary Tab. 1). Consistent with our findings, we found that survival rates were significantly lower in the PLK1-low compared with PLK1-high group. The Kaplan–Meier survival analysis revealed a 10-year survival rate of 33% for patients with low PLK1 expression and 82% for those with high PLK1 expression ($P = 0.011$).

**Discussion**

Chromosomal instability (CIN) is a hallmark of human neoplasms. The ability of tumors to adapt to external pressures is facilitated by tumor cell heterogeneity. Mechanisms responsible for this heterogeneity involve elevated frequencies of whole-chromosome missegregation—otherwise known as CIN. Chromosome missegregation is an important mechanism of tumor adaptation. Here, we studied the role of PLK1 in colon cancer cells with CIN by using the PLK1 inhibitor BI6727 (volasertib) or RNAi. The efficacy of this clinically most advanced PLK1 inhibitor has been demonstrated in different colon cancer models, and various phase I trials in solid tumors including colon cancer have shown favorable pharmacokinetics and manageable toxicity[52]. Moreover, the FDA has granted a Breakthrough Therapy designation to volasertib for the treatment of patients with AML. Volasertib, currently in Phase III clinical trials in combination with cytarabine, is reviewed as a promising agent for AML patients from the viewpoints of potential compliance and efficacy.

In this study, we used two colon cancer cell lines for the expression of truncated APC. In agreement with previous observations the expression of different APC mutants like the N-terminal 750 amino acids of APC (APC-ΔC) increased the phenotypical variability[34–36]. We demonstrate that the mitotic arrest induced by the PLK1 inhibitor BI6727 is attenuated in APC-ΔC-expressing colon cells. Our observations suggest that the

**Fig. 8** Inhibition of *Plk1* by RNAi or BI6727 increases the incidence of intestinal polyps in mice. **a** Scheme of the experimental procedure for the generation of transgenic mice. **b** Total cellular protein was prepared from the intestinal tissues of adult *Plk1-iKD* (±Dox) and separated by SDS–PAGE for anti-Plk1, -itetR, and -β-Actin immunoblotting analyses. **c** Hematoxylin/eosin-stained sections of normal and tumor tissues from $Plk1^{iKD}$, $Apc^{Min/+}$ $Plk1^{iKD}$ and $Apc^{Min/+}$ mice (±Dox) (×100). Sections were subjected to immunohistochemical evaluation after staining with IgGs to Ki-67 (×100). Black arrows highlight low grade (C,E) and high-grade (D,F) adenomas. (**d**) Mice of different genotypes and treatment schedules were sacrificed. Intestines from each mouse were examined using the Swiss role technique under a dissection microscope for tumor polyps. Average number of adenomas in colon and the small intestine from $Plk1^{iKD}$, $Apc^{Min/+}$ $Plk1^{iKD}$ with or without Dox-treatment, $Apc^{Min/+}$ with or without BI6727-treatment are shown. No visible tumor masses were detected in WT and $Plk1^{iKD}$ mice. **e** Sections of intestines from each mouse were examined under the microscope for the determination of the histological grade. Quantification of low and high grade adenomas in the small and large intestine from $Apc^{Min/+}$ $Plk1^{iKD}$ with or without Dox-treatment, $Apc^{Min/+}$ with or without BI6727-treatment are shown. **f** Since the expression of the Ki-67 protein is strictly associated with cell proliferation, we used the staining of Ki-67 in intestinal tissues to dermine the proliferative activity in corresponding sections. The staining was quantified based on different cutoff levels (<10%, ≤50% and >50%). (means ± s.d., $n = 3$, for each treatment). **P < 0.01, ***P < 0.001, Student's *t*-test, unpaired and two-tailed

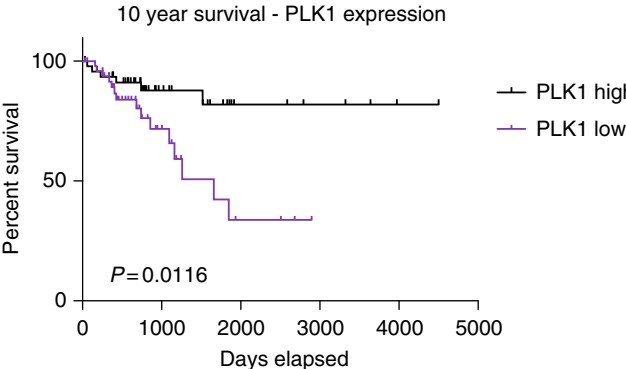

**Fig. 9** Overall survival of colon cancer patients according to PLK1 expression. Kaplan-Meier survival data (overall survival) for patients with colon cancer, separated by PLK1 expression (PLK1-high, PLK1-low groups are defined by median of FPKM values). (*p*-value of 0.011, based on log rank Mantell-Cox test, Graphpad Prism)

expression of APC-ΔC can compromise the SAC in colon cancer with different degrees of chromosomal instability. Several studies reported that manipulating APC function interferes with the mitotic checkpoint activity, which is consistent with an enhanced aneuploidy in these cells[50,53]. According to Dikovskaya et al. APC depletion by RNAi affected the SAC via downregulation of BUB1 at kinetochores concomitant with the reduction of BUBR1 at these sites[50]. Our data support this view by showing that expression of APC-ΔC weakens the SAC, as indicated by the reduced localization of BUBR1 and MAD1 at kinetochores causing, at least in part, the enhanced aneuploidy in both cell lines. However, how exactly APC modulates SAC activity or how it regulates BUB1 and BUBR1 functions still remain elusive. Intriguingly, we observed in our IF analyses a reduction of the Aurora B signal at kinetochores, 26% in APC-ΔC-expressing HCT116 cells and 30% in APC-ΔC-expressing SW480 cells, respectively along with a slight decrease in its activity compared to control cells. Considering the kinetochore localization of APC during mitosis and its role in stabilizing microtubule-kinetochore attachments, the expression of APC-ΔC might affect the structural integrity of kinetochores that represents the recruitment platform for other mitotic proteins fulfilling several mitotic activities (Supplementary Fig. 6 and Supplementary Fig. 9). Still, additional experiments are required to shed more light on kinetochore function and kinetochore-MT dynamics in cells expressing APC-ΔC.

Our work showed that inhibiting PLK1 activity in APC-ΔC-expressing cells lead to an extensive reduction of Aurora B at the kinetochores: 49% in HCT116 (APC-ΔC, BI6727) and 42% in SW480 (APC-ΔC, BI6727). This delocalization causes a dramatic loss of its activity at these sites. This is consistent with studies showing that PLK1, through the activation of Haspin kinase and the successive phosphorylation of Histone 3 on Thr 3 (H3 pT3), influences the recruitment of the CPC to kinetochores[48,49]. However, von Schubert et al. reported a normal recruitment of Aurora B in RPE-1 cells that were arrested in metaphase using the proteasome inhibitor MG132 and treated only for 1 h with the PLK1 inhibitor TAL[54]. During the MG132-induced metaphase arrest, the kinetochore recruitment of Aurora B is already established and involves additional mechanisms, such as, the phosphorylation of Histone 2A (T120) by Bub1 kinase[55]. In our experiments, the cells underwent a prolonged incubation with BI6727 (16 h) and therefore entered the G2 phase and mitosis with reduced PLK1 activity. This procedure allowed us to detect a significant reduction of Aurora B at kinetochore, which likely accounts for the early inhibition of H3T3 phosphorylation. Thus,

different protocols of drug exposure and different efficacies of the PLK1 inhibitors TAL and BI6727 may explain the discrepancies between our results and previous observations. Moreover, based on our data we can postulate that there is a requirement of PLK1 activity at early phases of mitosis in order to achieve a proper recruitment of Aurora B to kinetochores. During checkpoint initiation PLK1 phosphorylates in vitro and in vivo several MELT motifs on KNL1 protein that are required for the recruitment of BUB3:BUB1 and BUB3:BUBR1 to kinetochores[54,56]. This is in agreement with the results of our study, as the inhibition of PLK1 in APC-ΔC cells lead to a significant reduction of BubR1 intensity at kinetochores and to a strong reduction of the mitotic checkpoint complex (MCC) assembly. Thus, the loss of PLK1 activity in APC-ΔC-expressing cells with a compromised SAC, revealed the key role of PLK1 for the recruitment of checkpoint components and for the regulation of Aurora B localization and activity at kinetochores. Furthermore, the recruitment of HEC1 in APC-ΔC-expressing cells treated with BI6727 seems to be impaired. As HEC1 is critical for the kinetochore localization of MPS1, a reduced amount and activity of this important kinase at kinetochores of BI6727-treated APC-ΔC cells would not be surprising. This in turn will likely cripple even more the SAC efficiency in these cells. To address this aspect in more detail, additional experiments are required.

Our experiments indicate that while PLK1 inhibition leads to a partial impairment of Aurora B function, the use of Aurora B inhibitors has a more pronounced effect on polyploidy and colony formation of colonies of APC-ΔC-expressing cells. According to our data, the losses of PLK1 and APC functions cooperate to silence the SAC promoting a premature mitotic exit. All these factors provide a plausible explanation to the increased mitotic slippage and aneuploidy observed in APC-ΔC cells upon incubation with BI6727 (Supplementary Fig. 11).

Moreover, we have demonstrated that APC-ΔC-expression improves survival of BI6727-treated cells and increases CIN in surviving cells (Figs 3–5). To elucidate whether our observations have an impact on intestinal tumorigenesis in animals, we have analyzed two independent mouse models (Fig. 8). Despite many trials using Xenograft models for the evaluation of *Plk1* in cancer development[57–59], to our knowledge we have analyzed here for the first time the role of *Plk1* in a transgenic, murine cancer model. Initially we treated *Apc*$^{Min/+}$ mice with BI6727. Moreover, we generated *Apc*$^{Min/+}$ *Plk1*$^{iKD}$ mice using our previously devised murine inducible knock-down model *Plk1*$^{iKD}$ [32]. Remarkably, in both models inhibition of Plk1 increased the number of adenomatous polyps. The adenomas in Dox-treated *Apc*$^{Min/+}$ *Plk1*$^{iKD}$ mice and in BI6727-trated *Apc*$^{Min/+}$ mice were of higher grades than those observed in *Apc*$^{Min/+}$ mice. Our current investigation suggests that both, *Apc* and *Plk1*, genes play an important role in the suppression of intestinal carcinogenesis. Additional evidence for a tumor-suppressive role of *Plk1* came from experiments showing that heterozygous knockout mice developed tumors at a 3-fold-higher frequency than wild-type mice, confirming a role of *Plk1* in early development and tumor suppression[60].

In colon cancer patients harboring a nonsense mutation within the APC gene which leads to the expression of a truncated APC protein, high PLK1 expression increased patients's survival significantly (Fig. 9). Our data relate to recent findings about PLK1 in breast cancer cells showing that PLK1 modulates estrogen-dependent transcription of genes that have important tumor suppressive functions[61]. For another member of the PLK family, PLK4, was demonstrated that haploinsufficient *Plk4 + /-* mice have a higher incidence of liver and lung cancer, which was due to an impaired regulation of Cyclins D1, E and B1 and of CDK1 supporting a critical role of polo-like kinases for tissue homeostasis[62].

The number of PLK1 inhibitors that enter the clinic is still increasing and therefore it is crucial to improve our knowledge for the optimal clinical application. Cancer tissues with both P53 deficiency and/or RAS mutations and high PLK1 expression may be particularly sensitive to PLK1 inhibitors[63,64]. In line with these observations, cancer cells with wild-type P53 were shown to be less sensitive to the loss of PLK1 activity than P53-deficient cells[65]. However, recent study suggests the effectiveness of PLK1 inhibitors may depend also on the status of P53 functions in the affected cells[66]. In the context of our observations, preclinical findings suggest that PLK1 could be a particularly relevant target for cancers characterized by certain cancer-associated mutations (P53, RAS and APC). These genetic alterations could be important when establishing patient selection guidelines for anti-PLK1 therapy. We believe that this work improves our understanding of how PLK1 inhibition can influence the fate of genetically instable colon cancer cells and might provide a rational to strengthen the SAC in aneuploid tumors for cancer treatment.

## Methods

**Cell culture**. HCT116 was maintained in MacCoy's 5a and SW480 cells in RPMI 1640. Both cell lines were obtained from ATCC. Media were supplemented with 10% fetal calf serum, 100 U/ml penicillin and 100 μg/ml streptomycin (GIBCO and PAA) at 37 °C with 5% $CO_2$ in a humidified atmosphere. MEFs were prepared from 15-day-old embryos by a standard procedure[32]. The Eg5 inhibitor STLC was obtained from Tocris and BI6727 from Selleckchem.

**Cloning and generation of stable APC-ΔC-expressing clones**. APC-ΔC was inserted into the 3xFlag-tagged pcDNA3.1-Hygro + (Invitrogen) vector via standard cloning using 5′-ACTGGATCCATGGCTGCAGCTTCAT ATGA-3′ as forward and 5′-ACTGCGGCCGCTTATCTATCTTTTTCAGAACGAGAACTATC-3′ as reverse primer, respectively. HCT and SW480 cells were transfected with Flag- or APC-ΔC-expressing vectors and subjected to Hygromycin selection for the generation of stable clones.

**Western blot**. Cells were transfected with empty Flag- or Flag APC-ΔC plasmids and treated with STLC or BI6727 as indicated. After the incubation cells were lysed with RIPA buffer (50 mM Tris-HCl pH 8.0, 150 mM NaCl, 1% Triton X-100, 0.5% Na-desoxycholate, 0.1 % SDS, 1 mM $Na_3VO_4$, 1 mM PMSF, 1 mM DTT NaF and protease inhibitor complete (Roche). Polyvinylidene difluoride (PVDF) membranes were used for Western blotting applications. Blocking of membranes was performed with TBST including 2% BSA. The following antibodies were used at the indicated concentrations: mouse monoclonal PLK1 (F-8:sc-17783) (1:1000), MAD2 (17D10: sc-47747) (1:1000), BUB3 (31:sc-136217) (1:1000), CDC20 (H-175: sc-8358) (1:1000), BUBR1 (C-20:sc-16195) (1:1000), c-MYC (N-262:sc-764) (1:1000) (all Santa Cruz, Biotechnology Heidelberg, Germany), β-actin (AC-15: A1978) (1:200.000), Flag (M2:A8592) (1:1000), (Sigma-Aldrich, Taufkirchen, Germany), Cyclin D1 (1:1000) (EPR2241#2261−1) (Epitomics), APC (FE9# OP44) (1:1000) (Calbiochem, Darmstadt, Germany), Cyclin B1 (#4138) (1:1000), Aurora B (#3094) (1:1000), CENPA (Ser7 #2187) (1.1000), β-Catenin (D10A8 #8480) (1:1000) (Cell Signaling), pAurora B (T232 #ab61074) (1:1000) (Abcam), rabbit anti-phospho-Histone H3 (S10 #06-570) (1:1000) (Millipore, Schwalbach, Germany), HRP-conjugated secondary antibodies (1:5000) (GE Healthcare), itet-R (#TET01) (1:1000) (MoBiTec, Göttingen). The ECL Western Blotting Substrate was used for detection.

**Cell cycle and cell viability assays**. For cell cycle analysis, cells were harvested, washed with PBS, fixed in chilled 70% ethanol at 4 °C for 30 min, treated with 1 mg/ml RNase A (Sigma-Aldrich), and stained with 100 μg/ml propidium iodine for 30 min. Cell cycle quantification was performed using a FACS Calibur instrument and Cellquest Pro software (both BD Biosciences).

Caspase-Glo 3/7 assay kit (Promega) was used according to the manufacturer's instructions. The measured Luminescence (RLU) was presented as the mean value ± s.d. ($n = 3$). Apoptotic loss of membrane asymmetry was analyzed by staining for PE Annexin V and 7-AAD (BD Biosciences) and quantified on a FACS Calibur instrument. For apoptosis assays, non-synchronized cells were treated with the test compounds for the time indicated. Cell viability assays were conducted using the Cell Titer-Blue Cell Viability Assay (Promega) according to the manufacturer's instructions using fluorescence as a read-out (excitation/emission wavelengths: 562 nm/615 nm). The significance of differences between populations of data was assessed according to the Student's two-tailed test (*$P ≤ 0.05$; **$P ≤ 0.01$; ***$P ≤ 0.001$).

**Immunofluorescence microscopy and antibodies for immunofluorescence**. For indirect immunofluorescence staining cells were seeded on cover slides. Briefly, cells were treated for the indicated time points, then fixed for 5 min in −20 °C methanol and permeabilized for 10 min at room temperature with 0.1% Triton X-100. The following primary antibodies were used for staining: Aurora B (6/AIM-1 # 611082) (BD Biosciences), BUBR1 (8G1#ab4637) (Abcam), CASC5 (#ab70537) (Abcam), NDC80 (9G3.23) (Novus Biologicals), ZWINT-1 (IHC-00095) (Bethyl laboratories), CENP-A pS7 (NL41#04-792) (Millipore), Histone 3 pT3 (#06-570) (Millipore), MAD1 (#ab175245) (Abcam), and human immune serum against centromere (anti-centromere antibody, ACA, ImmunoVision, springdale, USA). FITC, Cy5 and Cy3 conjugated secondary antibodies were obtained from Jackson Immunoresearch (Newmarket, UK). DNA was stained with DAPI (Roche). Images were taken using an AxioObserver.Z1 microscope with a HCX PL APO CS 63.0 × 1.4 oil UV objective (Zeiss, Göttingen) and a confocal laser-scanning microscope (CLSM, Leica CTR 6500, Heidelberg). Images were imported in the software ImageJ Fiji, which was used to measure the kinetochore intensities.

**Time lapse microscopy**. Thymidine-synchronized HCT116 cells expressing mCherry-histone H2B were released for 5 h, treated either with 100 nM BI6727 or 2 μM of the Eg5 inhibitor STLC. For time-lapse analysis, the treated cells were transferred to the microscope stage and microscopy was performed with Axioimager inverted Z1 (Zeiss) equipped with an environmental chamber (Zeiss) that maintained the cells at 37 °C in a humidified environment of 5% $CO_2$. Images were taken every 10 min using an Axiocam MRm camera (Zeiss) driven by Axiovision SE64 software (Zeiss). Movies and JPEG files were imported into ImageJ and proceeded using the same software. Nuclear envelope breakdown was judged as such when the nuclear membrane lost a smooth and the linear periphery. The first frame showing a pole ward movement of the chromosomes was defined as anaphase onset.

**Chromosome spreads**. Cells were treated overnight with 3.3 μM Nocodazol. The next day cells were harvested by mitotic shake off and hypotonically swollen in 40% medium/ 60% tap water for 20 min at 37 °C. Cells were fixed with freshly made Carnoy's solution (75% methanol, 25% acetic acid), and the fixative was changed several times. For spreading, cells in Carnoy's solution were dropped onto pre-chilled glass slides. Slides were dried at room temperature for 24 h and stained with DAPI. Chromosome number per condition was counted using an AxioObserver.Z1 microscope with a HCX PL APO CS 63.0 × 1.4 oil UV objective (Zeiss, Göttingen). The graphic representation of the results was done using GraphPad Prism software.

**In vitro fertilization and generation of Apc^Min/+ Plk1^iKD mice**. Mice were kept in the animal facility at Taconic Artemis GmbH in microisolator cages (TecniplastSealsave, Hohenpeißenberg). The Apc/Plk1 colony started out as breeding at Taconic. Two separate lines Apc^Min/+ and Plk1^iKD were setup for breeding to generate double heterozygous males to be used for rapid expansion via in vitro fertilization. The IVF was performed by taking sperm from the double heterozygous Apc^Min/+ Plk1^iKD males and oocytes from C57BL/6NTac females, mixing them and incubating. Once the embryos reached 2-cell stage they were implanted into our germ-free recipient females who carried out the rest of the pregnancy. After health clearance, all animals were shipped back to the animal facility of the Medical School (Goethe University, Frankfurt). All animal studies were approved according to the German Animal Welfare Act according to the permission (14.5.2013) by the Regierungspräsidium Darmstadt. Animals were randomized into treatment and control groups of ten mice each. The Dox-treatment was performed as described previously[67]. In brief, the drinking water which contained 2 mg/ml Dox (Sigma, München) and 10% sucrose was prepared every other day (kept in the dark). For the oral treatment with the small molecule inhibitor, BI6727 was resuspended in 0.5% Natrosol 250 hydroxyethyl-cellulose and given intragastrally via gavage needle. An administration volume of 200 μl BI6727 of a stock solution (5 mg/ml)was used to obtain a dose of 10 mg BI6727/kg body weight. The use of animals complied with the regulations of the ZFE (Goethe University, Frankfurt). Details of the animal experiments that may influence results were included in the manuscript based on the ARRIVE reporting guidelines for the documentation of animal trails[68].

**Analysis of mouse genotypes**. To test the genotype of wt or transgenic mice, genomic DNA was prepared from tail clips 0.5–0.8 mm in length with Viagen Direct PCR-Tail reagent (Peqlab Biotechnologie, Erlangen) according to the manufacturer's protocol. For the standard PCR, 10 ng genomic DNA was amplified using the sense primer 5′- ATCGCGGGCCCAGTGTCACTAGGC-3′ and the antisense primer 5′-CTAGTACGCGCCTGCAGGCTAGCC-3′ to amplify the Plk1^iKD cassette. The original plasmid containing the cDNA of the KD cassette served as the positive control. For the amplication of the APC gene the following primers were used: 5′-GCCATCCCTTCACGTTAG-3′, 5′-TTCCACTTTGGCAT AAGGC-3′ and 5′-TTCTGAGAAAGACAGAAGTTA-3′.

**Pathology screen**. Mice were sacrificed with $CO_2$ and analyzed macroscopically and weighed. The small and large intestines were removed from age-matched mice for histological, immunohistochemical and western blot characterization of the intestinal tract. Isolated small and large intestines were flushed with modified Bouin's fixative (50% ethanol, 5% acetic acid, and 10% formaldehyde), and cut open longitudinally for gross examination. The intestines were then rolled into a Swiss-roll, fixed in 4% buffered formalin, and embedded in paraffin. Two-micrometer sections were cut and stained with hematoxylin and eosin. Age-matched control and mutant mice were examined histologically for the number and location of the tumors.

For immunohistochemistry, air-dried sections from formalin-fixed, paraffin-embedded intestinal tissue were pretreated with trilogy solution (cell marque) for 30 min in a water bath for antigen retrieval and then stained with polyclonal primary antibody to Ki67/MKI67 (NB600-1209, Novus Biologicals). As internal positive controls served the lymphatic tissue within the intestine mucosa. Sections without application of the primary antibody were used as negative controls.

**Bioinformatics analysis methods**. Colon cancer data were generated by the TCGA Research Network (http://cancergenome.nih.gov/) was downloaded from the Genomic Data Commons repository through TCGABiolinks[69] R package.

SNP calls from MuSE, MuTect, VarScan and SomaticSniper were combined and filtered for nonsense mutations in the gene APC. (A patient was selected if at least 1 algorithm indicated that APC gene had a nonsense mutation.) Expression values for PLK1 and clinical data for these samples were extracted and used for the survival analysis. The samples were split into high and low groups using PLK1 expression. Survival curves and the associated log-rank p-value was obtained using the TCGAanalyze_survival function of TCGAbiolinks package[70].

FPKM and upper quantile normalized FPKM values of PLK1 was split into high/low groups using mean, median and percentile combinations of 60–40 (higher than 60% and lower than 40%), 75–25 and 90–10. The only combination that provided an uncorrected significant p-value ($p < 0.05$) was upper quantile normalized FPKM using median PLK1 expression in APC mutated samples.

**Statistical methods**. All experiments were performed at least in triplicate. Standardization and statistics were determined as described[15]. In brief, statistical analysis was performed using Microsoft Excel and GraphPad Prism software. For paired t-tests, all experimental groups were compared with their respective groups. Student's t-test was used to determine statistical significance between two groups. Significant differences (*$P \leq 0.05$; **$P \leq 0.01$; ***$P \leq 0.001$) are indicated in the figures with asterisks. The Mann–Whitney U-test was used as a non-parametric test that is used to compare two sample means that come from the same population, and used to test whether two sample means are equal or not.

**Data availability**. The data that support the findings of this study are available within this article and from the corresponding author upon reasonable request.

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

## Acknowledgements

We thank Eduardo Augusto Alonso for supporting the animal experiments. This work was supported by grants from the Wilhelm Sander-Stiftung, the German Cancer Consortium (DKTK), (Heidelberg), Deutsche Krebshilfe, the Carls-Stiftung and the Research Support Foundation.

## Author contributions

M.R., M.S. and K.S. conceptualized, coordinated study and helped to write the manuscript. M.R., M.S., Y.M. designed and performed experiments, interpreted results and prepared figures. A.H. performed imaging and pathological examination, I.L., C.D., and E.K.-C. performed cell culture experiments, N.H. and O.W. designed and performed animal studies, R.F. performed the Kaplan–Meier anlysis, and S.B. helped to coordinated study and in the writing of the paper.

## Additional information

**Competing interests:** The authors declare no competing interests.

