## [Peer Review File(PDF 619 kb) · Nature Communications]

Reviewers' comments:

Reviewer #1 (Remarks to the Author):

Nature Comm

Plk1 has tumor-suppressive potential in APC-truncated colon cancer cells

It is generally accepted that overexpression of Polo-like kinase 1 (Plk1) promotes the development of various cancers in humans. However, sporadic evidence suggests that down-regulation of Plk1 also induces tumorigenesis (Lu, LY et al, 2008), although how this apparently paradoxical process can occur has remained largely elusive. In this manuscript, the authors showed that upon Plk1 inhibition, expression of an Adenomatous polyposis coli (APC)-truncated mutant (Δ APC), lacking the N-terminal 750 amino acids, induced premature mitotic exit and increased the survival of cells with chromosomal abnormalities. These cells exhibited a diminished level of Aurora B at kinetochores and compromised the formation of the mitotic checkpoint complex (MCC). Consistent with these findings, Plk1 depletion (in Plk1-iKD mice) promoted the development of adenomatous polyps in APC^{min/+} mice, a well characterized colorectal cancer model. Furthermore, a high level of Plk1 expression appeared to correlate with the better survival rate of colon cancer patients expressing truncated APC mutants. Based on these data, the authors suggest a potential role of Plk1 as a tumor suppressor in APC-truncated colon cancer.

Overall, the main conclusion of the manuscript suggesting that Plk1 inhibition augments the tumorigenic capacity of a dominant-negative Δ APC mutant and potentially worsens the clinical outcome of APC-defective colon cancers appears to be solid. This study may have identified another genetic element that may play an important role in determining the efficacy of anti-Plk1 therapy in clinical settings.

Major concerns:

1. It has been shown that patients with familial adenomatous polyposis have germline mutations mostly leading to premature translational termination, while over 60% of somatic APC mutations occur within a mutation cluster region (aa 1250-1500). In this study, the authors used a system that overexpresses a dominant-negative Δ APC mutant lacking the N-terminal 750 residues. Therefore, it is not clear whether Plk1 inhibition may induce the same effect as shown in this study, when it is combined with APC RNAi or other commonly occurring cancer-associated APC mutations. It would be very helpful, if this can be addressed. The authors should also provide an explanation as to why they chose the APC Δ (1-750) mutant over other cancer-associated mutations.
2. Inactivation of APC and activation of the beta-catenin pathway are commonly associated with the development of colon cancer. The authors stated that the Δ APC mutant lacking its N-terminal 750 amino acids is devoid of all the beta-catenin regulatory sequences (line 117). It is well appreciated that deregulated beta-catenin activity results in uncontrolled cell growth that may lead to cancerous tumors. Therefore, it is somewhat surprising that the authors did not attempt to address whether and, if so, how Plk1 inhibition alters (or cooperates with) an APC-dependent beta-catenin-mediated pathway(s), and whether this alteration (or cooperation) can be accountable for the enhanced tumorigenesis observed in Plk1 inhibitor-treated Δ APC-expressing cells.
3. Fig. 7 and Suppl Fig. 6 - It has been suggested that APC-deficient cells have a compromised SAC function (Radulescu S, et al., Oncogene 2010). Interestingly, the authors found that expression of Δ APC alone was sufficient to diminish the level of BubR1 (Fig. 7) and inhibit the formation of the mitotic checkpoint complex (MCC) (Supl Fig. 6). In addition, Plk1 inhibition in the Δ APC-expressing cells, but not the control cells, exhibited a defect in assembling MCC (Fig. 7). Based on these observations, the authors suggest that cells expressing the Δ APC mutant may have a defect in the structural integrity of kinetochores, and that Plk1 inhibition may further

cripple the SAC function. Further investigation on the molecular mechanism of how the effect of the dominant-negative Δ APC mutant can be potentiated by Plk1 inhibition will be very important. Is the effect of the Δ APC mutation and Plk1 inhibition in Fig. 7 and Suppl Fig. 6 additive? It looks like this could be the case, but not clearly stated.

4 (Continuation of comment #3). If Plk1 inhibition-induced SAC defect is the primary reason for the increased cell survival (Fig. 4) and adenomatous polyp formation (Fig. 8), then inhibition of AurB or Haspin may also induce a similar effect in Δ APC-expressing cells (as postulated in line 429). If this is the case, it will greatly strengthen the author's model shown in Supp Fig. 8 that Plk1 and APC cooperate (or coordinate) their activities to assemble MCC and establish SAC.

5. The authors heavily rely on BI6727 to inhibit Plk1. Although BI6727 is the most advanced anti-Plk1 inhibitor at present, it still exhibits considerable cross-reactivities against other kinases. The authors should use Plk1 RNAi to verify the effect of BI6727 in Δ APC-expressing cells.

6. A dramatically increased aneuploidy in Plk1 inhibitor-treated Δ APC-expressing cells (Figure 5) and an equally impressive effect on the formation of polyps in APCmin/Plk iKD mice (Figure 8) are quite striking.

7. It has been suggested that cancer cells with an inactivated TP53 or activated RAS mutation are more sensitive to Plk1 inhibition than their isogenic wild-type cells. Given the importance of genetic variations in determining the efficacy of anti-Plk1 therapy, the effect of the Δ APC mutation on Plk1 inhibition described in this study suggests that profiling APC mutations could be important when establishing patient selection guidelines for anti-Plk1 therapy. Adding a paragraph discussing this point may help highlight the main finding of this work.

Minor comments:

1. Results shown in Fig. 1 and 2 are somewhat redundant, and can be presented more concisely. Perhaps, one of them could be placed in the supplementary data.
2. Fig. 1 legend - Please clarify what "UT" stands for. It looks like it is shorted for "untreated".
3. Page 6 - need to rewrite "two enrichment steps ---". It appears that the two enrichment steps are 1) Bi6727 treatment and 2) mitotic shake-off. It is not clearly written.
4. Supplementary Fig. 7 - More explanation for all the samples would be needed for readers to better understand the data provided here.

Reviewer #2 (Remarks to the Author):

This is an interesting study on the combined effects in cells of APC truncation and Plk1 kinase inhibition. Aurora B kinase localization to kinetochores and spindle assembly checkpoint (SAC) arrest are perturbed leading to aneuploidy.

In addition, Plk1 inhibition promotes the development of polyps in mouse models (APC Min/+) and Plk1 expression levels are shown to correlate with survival of colon cancer patients.

There are a number of issues that the authors need to address here before publication:

Figure 1

The reader should be given a clearer description of the APC genotype in the different cells. In SW480 cells, are both APC alleles truncated?

Ideally I would like to see data generated from live-cell imaging (time-lapse movies) for certain experiments performed in Figs 1-3. Have any movies been made, to corroborate the fixed time point data shown here?

Whilst the trends are clear, the statistical significance of the data here isn't always convincing. For example, the effect after 3 days of Plo1 inhibition (lines 129-130 in the text): this shows a reduction of G2/M cells from 61% to 45%. After 2 days the difference was from 69 to 57%. These drops seem relatively subtle, over such long time courses.

The scoring of cell cycle stages, from the FACS data, needs to be more carefully explained in the Figure legends.

Figure 2

Why do the SW480 cells exit mitosis more slowly than the HCT116 cells, after their release from Plk1 inhibition. There is a difference of several hours, yet these are the cells that have mutated APC and the CIN phenotype. The authors need to explain and discuss these differences in timing.

Figure 3

It is argued that inhibition of Plk1 and Delta APC lead to endoreduplication (line 200). How do the authors know this, rather than these cells having failed cytokinesis, possibly due to defects in chromosomal passenger complex functions. Have cells been analyzed whilst undergoing cytokinesis – live-cell imaging would help here.

Figure 4

More explanation of these assays is needed, for the general reader. Figure legends are too brief. For example, what should we take away from the phase-contrast images (which are rather small) and the coomassie stained dishes? The 'MTT assay' needs a brief explanation here too.

Figure 6

The authors really need to look at Mps1 kinase localization and its levels at kinetochores. Surprisingly this key checkpoint kinase is rarely mentioned, and it almost certainly has key role(s) here:

- along with Plo1 it targets KNL1 for phosphorylation (and thereby Bub1/BubR1 recruitment to kinetochores)
- it also phosphorylates Bub1, leading to Mad1 recruitment to kinetochores

Mps1 kinase activity is likely co-ordinated with (and/or regulated by) Aurora B kinase activity, so Mps1 levels may well be down at these kinetochores.

This study involves direct perturbation of Plk1 activity, but Aurora B kinase and Mps1 kinase are being indirectly affected (deltaAPC) – it is not surprising that mitosis and chromosome segregation are significantly perturbed in these cell lines.

Figure 7

Again Mps1 is a key regulator of MCC assembly, and this needs to be discussed here. MCC levels will drop as cells exit mitosis, so it is not clear here if the low levels are a cause or consequence of slippage. The authors need to block proteasome activity and repeat this experiment: are MCC levels still low?

Figure 8

Ki67 needs better introduction here, as does generation of the plot in 8f. We can't see the Ki67 staining data clearly (Fig 8c).

Reviewer #3 (Remarks to the Author):

Within the manuscript entitled 'Plk1 has tumour-suppressive potential in APC-truncated colon cancer cells' the authors major claim is that Plk1 along with APC play an important role in suppressing tumorigenesis and upon mutation of APC, inhibition of Plk1 results in silencing of the SAC so promoting a premature exit from the cell cycle resulting in an increase in aneuploidy. The cell culture and in vivo models used in this manuscript show that inhibiting or knocking down Plk1 in conjunction with truncated APC results in an increase in cell division, polyploidy and small intestinal tumours which correlate with human data where CRC patients with low Plk1 have a poor survival. This suggests that any inhibitors of Plk1 would not have a clinical benefit in a CRC cohort. The role of Plk1 in tumorigenesis has previously been discussed where it was shown that Plk1-/+ mice developed spontaneous tumours more rapidly than wild-type counterparts. However, no previous work has focussed on the role of Plk1 following the loss of APC. Expression of truncated APC in colon cancer cells overrides partially the mitotic arrest induced by Plk1 inhibition. Overall there are some interesting points to the manuscript however there are many issues with the paper (detailed below, not least with data presentation and analysis) which preclude publication.

Specific comments

1. Throughout the paper, the authors use the small molecule BI6727 as an inhibitor of Plk1, however, they show in figure 1 that this protein is still present (and increases in expression following treatment with BI6727). They should demonstrate the effect of this inhibitor on the activity of Plk1/its downstream effector (or referencing how it works).

2. Members of the Plk family are well known regulators of cell cycle progression including mitosis and it is established that PLK function is associated with cancer¹. However, it is not clear if activity of Plk1 is altered upon loss of function of APC. From figure 1 it appears that expression of Plk1 is unaltered following expression of truncated APC. Does the activity of Plk1 change following expression of truncated APC. How is APC controlling Plk1 activity? Is it WNT dependent? The authors have not shown a direct link between Plk1 and APC – at present this link is assumed.

3. The exit of cells from a mitotic arrest induced by Plk1 inhibition is accelerated by truncated APC expression

The finding that loss of function of APC has no effect on proliferation of either cell line (figure S2) is perhaps surprising given that it is well established that loss of APC results in an increased proliferation in vivo². It would be interesting to see analysis of downstream WNT targets to determine how the truncated APC transgene is affecting well known pathways.

4. Impact of truncated APC expression on polyploidy and survival of Plk1 inhibitor-treated colon cancer cells

Throughout the manuscript the quality of images is very poor making it difficult to interpret some data. The text surrounding figure 4 talks about crystal violet staining, where the figure legend mentions coomassie staining – this should be consistent. The authors state that more floating cells and debris was observed in the truncated-APC cultures however, no data was shown to demonstrate this (this may be present in the figures, however due to the quality –I cannot see this). It is stated in the text that more caspase and annexin was observed in the controls vs APC counterparts at a range of concentrations, however the data show that at 24 hours more caspase was observed in the truncated APC cell lines. From the proliferation curves, it is clear that inhibition of Plk1 in cell expressing truncated APC results in an increase in cell proliferation.

5. The inhibition of Plk1 in truncated APC-expression cells increases chromosome abnormalities
Figure 5 again has poor quality images of the metaphase spreads (with no scale bar on fig S4a). In addition to this, the axis on the chromosome number histograms are all inconsistent making it difficult to compare and interpret. However, it is clear from the analysis shown in figure 5c that

chromosome number/cell increases when Plk1 is inhibited in cells expressing truncated APC.

6. Inhibiting Plk1 in truncated APC expressing cells reduces the protein level and activity of Aurora B at Kinetochores

In figures 6, 7, S5 and S6 – there is no IF carried out on non-treated cells. How is the localisation of Aurora B, H3 pT3, CENP-A P27 and BubR1 affected by BI6727 and loss of function of APC?

Whilst it has been shown that inhibiting Plk1 and/or APC function reduces that amount of AuroraB, H3pT3, CENPA pS7, BubR1 and Mad1m, there is no data to show how these are altered from a control, untreated sample. Additionally – all of the plots have been scatter dot plots except the Figure 6c – is there a reason for this?

7. The inhibition of Plk1 increases the frequency of colonic tumours in APCMin/+ mice

The authors say that inhibition of Plk1 increases the frequency of colonic tumours in APCmin mice. However, they do not show that in this manuscript. They do however, show an increase in tumour in the small intestine. Additionally, as with previous figures, the quality of the images is not sufficient to make any claims regarding the pathology of tumour. The authors claim that the tumours in the APCMin mice was invasive into the mucosa, however, this is not clear from the pictures provided. There are several claims regarding proliferation and Ki67 staining, however, from the images provided, I cannot see any Ki67 staining. Authors have used commas as opposed to decimal points on graphs. The authors do show that the number of small intestinal tumours are significantly increased following inhibition of Plk1 in an APCmin/+ model of CRC, which hasn't been shown previously.

8. Patients with colon tumours expressing low Plk1 have a worse clinical outcome

The authors show that in a human cohort of CRC patients, that high levels of Plk1 is significantly tumour suppressive (however the P value on the graph is incorrect). This is in contrast to a previous study where it has been shown that PLk1 is overexpressed in CRC cancer specimens and is linked to progression of a primary CRC tumour³.

9. Throughout the manuscript all statistical tests were carried out using students T-test, unpaired and two-tailed, which is not the best statistical test for some of the experiments (for instance in figures 3, 5, 6, 7) Mann-whintey would be more appropriate. Additionally, no exact P values were provided.

No catalogue or clone numbers for antibodies are provided, although suppliers are mentioned.

Overall, I think the authors have shown that inhibition of Plk1 in exacerbates the loss of function of APC phenotype particularly in the context of cell proliferation, aneuploidy and tumour formation, which has not been shown before. My concerns regarding this manuscript are that the figures are of very poor quality making it difficult to independently assess the data and no direct link between APC and Plk1 has been shown.

We are grateful for the valuable feedback provided by the reviewers in response to our manuscript entitled “Plk1 has tumor-suppressive potential in APC-truncated colon cancer cells”. In response to the concerns raised by the reviewers, we added novel data and made certain insertions and corrections to the manuscript, regarding both, the experimental work and the written material, while rebutting a few. Changes in the manuscript are underlined.

Reviewers' comments:

Reviewer #1 (Remarks to the Author):

Nature Comm

Plk1 has tumor-suppressive potential in APC-truncated colon cancer cells

It is generally accepted that overexpression of Polo-like kinase 1 (Plk1) promotes the development of various cancers in humans. However, sporadic evidence suggests that down-regulation of Plk1 also induces tumorigenesis (Lu, LY et al, 2008), although how this apparently paradoxical process can occur has remained largely elusive. In this manuscript, the authors showed that upon Plk1 inhibition, expression of an Adenomatous polyposis coli (APC)-truncated mutant (Δ APC), lacking the N-terminal 750 amino acids, induced premature mitotic exit and increased the survival of cells with chromosomal abnormalities. These cells exhibited a diminished level of Aurora B at kinetochores and compromised the formation of the mitotic checkpoint complex (MCC). Consistent with these findings, Plk1 depletion (in Plk1-iKD mice) promoted the development of adenomatous polyps in APC^{min/+} mice, a well characterized colorectal cancer model. Furthermore, a high level of Plk1 expression appeared to correlate with the better survival rate of colon cancer patients expressing truncated APC mutants. Based on these data, the authors suggest a potential role of Plk1 as a tumor suppressor in APC-truncated colon cancer.

Overall, the main conclusion of the manuscript suggesting that Plk1 inhibition augments the tumorigenic capacity of a dominant-negative Δ APC mutant and potentially worsens the clinical outcome of APC-defective colon cancers appears to be solid. This study may have identified another genetic element that may play an important role in determining the efficacy of anti-Plk1 therapy in clinical settings.

Major concerns:

1. It has been shown that patients with familial adenomatous polyposis have germline mutations mostly leading to premature translational termination, while over 60% of somatic APC mutations occur within a mutation cluster region (aa 1250-1500). In this study, the authors used a system that overexpresses a dominant-negative Δ APC mutant lacking the N-terminal 750 residues. Therefore, it is not clear whether Plk1 inhibition may induce the same effect as shown in this study, when it is combined with APC RNAi or other commonly occurring cancer-associated APC mutations. It would be very

helpful, if this can be addressed. The authors should also provide an explanation as to why they chose the APC Δ (1-750) mutant over other cancer-associated mutations.

We are sorry, but this might be a misunderstanding. At the beginning of the result section on page 5 (1. paragraph) the manuscript says: “To investigate the role of Plk1 in cancer cells with CIN, we expressed in both cell lines a truncated APC protein encompassing the N-terminal 750 amino acids of APC (Δ APC).” i.e. the APC protein that was expressed in all experiments contains the first 750 aa of APC. To improve this statement, we have reworded this sentence on page 5.

According to Rowan et al. (doi: 10.1073/pnas.97.7.3352) there is a deficiency of APC mutations before codon 750 in the sporadic colorectal cancers i.e. the majority of mutations were detected C-terminal to codon 750 predominately in the MCR (codons 1,286–1,513). To reflect the majority of those mutations in sporadic cancer, we expressed the first 750 aa in the colorectal cancer cell lines tested.

2. Inactivation of APC and activation of the beta-catenin pathway are commonly associated with the development of colon cancer. The authors stated that the Δ APC statement mutant lacking its N-terminal 750 amino acids is devoid of all the beta-catenin regulatory sequences (line 117). It is well appreciated that deregulated beta-catenin activity results in uncontrolled cell growth that may lead to cancerous tumors. Therefore, it is somewhat surprising that the authors did not attempt to address whether and, if so, how Plk1 inhibition alters (or cooperates with) an APC-dependent beta-catenin-mediated pathway(s), and whether this alteration (or cooperation) can be accountable for the enhanced tumorigenesis observed in Plk1 inhibitor-treated Δ APC-expressing cells.

To address this aspect, we have expressed Δ APC in SW480 and in HCT-116 cells and observed an upregulation of the level of β -catenin. This upregulation was more pronounced in HCT116 cells that have two wild-type APC alleles. In an independent experiment we inhibited Plk1 function by using BI6727 or Plk1-specific siRNA for 24 h. This inhibition did not alter the level of β -catenin confirming previous data by Mbom et al. (Mol Biol Cell. 2014 Apr;25(7):977-91). The data indicate that an inhibition of Plk1 does not stabilize β -catenin. We added the new data to Fig. 4 and new text to page 9, 2nd paragraph.

3. Fig. 7 and Suppl Fig. 6 - It has been suggested that APC-deficient cells have a compromised SAC function (Radulescu S, et al., Oncogene 2010). Interestingly, the authors found that expression of Δ APC alone was sufficient to diminish the level of BubR1 (Fig. 7) and inhibit the formation of the mitotic checkpoint complex (MCC) (Supl Fig. 6). In addition, Plk1 inhibition in the Δ APC-expressing cells, but not the

control cells, exhibited a defect in assembling MCC (Fig. 7). Based on these observations, the authors suggest that cells expressing the Δ APC mutant may have a defect in the structural integrity of kinetochores, and that Plk1 inhibition may further cripple the SAC function. Further investigation on the molecular mechanism of how the effect of the dominant-negative Δ APC mutant can be potentiated by Plk1 inhibition will be very important. Is the effect of the Δ APC mutation and Plk1 inhibition in Fig. 7 and Suppl Fig. 6 additive? It looks like this could be the case, but not clearly stated.

To address this point, we stained kinetochores of wild-type and Δ APC-expressing HCT116 cells for the kinetochore core components HEC1, KNL1, and ZWINT1 in the presence or absence of the Plk1 inhibitor BI6727. The analysis of the recruitment of these kinetochore components will deliver clues about the overall integrity of kinetochores. We could show that the only presence of Δ APC interferes with the structural integrity of kinetochore, as the recruitment of all markers was reduced. Furthermore, this recruitment was even more disrupted, when Δ APC expression was combined with Plk1 inhibition. This suggests that the loss of Plk1 activity and Δ APC expression act in an additive manner towards preventing a normal kinetochore assembly. This in turn leads to severe SAC dysfunction, as kinetochores represent the platform on which the spindle checkpoint components are assembled.

For the description of the novel data we added a new Suppl. Fig. 6, a description of the data on page 10-11, and added new text to the discussion on page 16, 1st paragraph.

4 (Continuation of comment #3). If Plk1 inhibition-induced SAC defect is the primary reason for the increased cell survival (Fig. 4) and adenomatous polyp formation (Fig. 8), then inhibition of AurB or Haspin may also induce a similar effect in Δ APC-expressing cells (as postulated in line 429). If this is the case, it will greatly strengthen the author's model shown in Supp Fig. 8 that Plk1 and APC cooperate (or coordinate) their activities to assemble MCC and establish SAC.

We placed several orders, but unfortunately, the Haspin inhibitors CHR 6494 and LDN 209929, are currently not available from commercial vendors. Due to this reason, we focused our efforts on the use of Barasertib (AZD1152-HQPA), which is a highly selective Aurora B inhibitor with IC50 of 0.37 nM in a cell-free assay, ~3700 fold more selective for Aurora B over Aurora A. To investigate whether the Plk1 inhibition-induced SAC defect is comparable to effects in Δ APC-expressing cells induced by the inhibition of Aurora B, we treated Δ APC-expressing and control cells with the small molecule Aurora B inhibitor AZD1152 to test for changes in proliferative activity, polyploidy and colony formation.

For the description of the novel data we added a new Suppl. Fig. 8, a description of the data on page 11, last paragraph and added new text to the discussion on page 16 (bottom).

5. The authors heavily rely on BI6727 to inhibit Plk1. Although BI6727 is the most advanced anti-Plk1 inhibitor at present, it still exhibits considerable cross-reactivities against other kinases. The authors should use Plk1 RNAi to verify the effect of BI6727 in Δ APC-expressing cells.

Recently, we performed a Kinobead competition assay to the quantitative profiling of the Plk1 inhibitors BI2536 and BI6727 (Raab et al., Cell research, 2014). Kinobeads captured approximately two-thirds of the expressed kinome. Our experiment based on a comprehensive mass spectrometry study revealed a very narrow target profile for BI6727 highlighting the specificity of the clinical inhibitor BI6727 (Raab et al., unpublished data). Still, according to the reviewer's suggestion we silenced Plk1 expression by RNAi and added the new data to Suppl. Fig. 2 (panel c) and new text to page 6 (4. paragraph).

Furthermore, the original manuscript describes animal experiments performed using the small molecule inhibitor BI6727 or RNAi (Fig. 8d, left and right panels).

6. A dramatically increased aneuploidy in Plk1 inhibitor-treated Δ APC-expressing cells (Figure 5) and an equally impressive effect on the formation of polyps in APC^{min}/Plk1KD mice (Figure 8) are quite striking.

7. It has been suggested that cancer cells with an inactivated TP53 or activated RAS mutation are more sensitive to Plk1 inhibition than their isogenic wild-type cells. Given the importance of genetic variations in determining the efficacy of anti-Plk1 therapy, the effect of the Δ APC mutation on Plk1 inhibition described in this study suggests that profiling APC mutations could be important when establishing patient selection guidelines for anti-Plk1 therapy. Adding a paragraph discussing this point may help highlight the main finding of this work.

We added new text to the discussion on page 17, last paragraph.

Minor comments:

1. Results shown in Fig. 1 and 2 are somewhat redundant, and can be presented more concisely. Perhaps, one of them could be placed in the supplementary data.

Old Fig. 2 has been placed into supplementary section (Suppl. Fig. 3) and replaced by the new time-lapse microscopy figure (new Fig. 2)

2. Fig. 1 legend - Please clarify what “UT” stands for. It looks like it is shorted for “untreated”.

Indeed, “UT” stands for untreated. We added this explanation to the legend of Fig. 1

3. Page 6 – need to rewrite “two enrichment steps ---“. It appears that the two enrichment steps are 1) Bi6727 treatment and 2) mitotic shake-off. It is not clearly written.

We reworded the corresponding text of page 7.

4. Supplementary Fig. 7 – More explanation for all the samples would be needed for readers to better understand the data provided here.

More specific information was added to the legend of Suppl. Fig. 10 (previously Suppl. Fig. 7) for a more detailed description of the samples.

Reviewer #2 (Remarks to the Author):

This is an interesting study on the combined effects in cells of APC truncation and Plk1 kinase inhibition. Aurora B kinase localization to kinetochores and spindle assembly checkpoint (SAC) arrest are perturbed leading to aneuploidy.

In addition, Plk1 inhibition promotes the development of polyps in mouse models (APC Min/+) and Plk1 expression levels are shown to correlate with survival of colon cancer patients.

There are a number of issues that the authors need to address here before publication:

Figure 1

The reader should be given a clearer description of the APC genotype in the different cells. In SW480 cells, are both APC alleles truncated?

In their study Rowan et al. (Proc Natl Acad Sci U S A. 2000 Mar 28;97(7):3352-7) screened the entire coding region of *APC* for mutations and assessed allelic loss in a set of 41 colorectal cancer cell lines including HCT-116 and SW480 cells. While both APC alleles were found to be non-mutated in HCT-116 cells, in SW480 cells one allele is mutated at codon 1,338 CAG to TAG and the second one is not mutated. Rowan et al. confirmed previous data published by Nishisho et al. (Science. 1991 Aug 9;253(5020):665-9).

We added more information to the genotypic description of the SW480 cell line on page 5, 1st paragraph.

Ideally I would like to see data generated from live-cell imaging (time-lapse movies) for certain experiments performed in Figs 1-3. Have any movies been made, to corroborate the fixed time point data shown here?

Live imaging of HCT116 and Δ APC-expressing cells in presence and absence of BI6727 has been generated and the results are represented in Figure 2. The description of the new data is provided on page 7.

Whilst the trends are clear, the statistical significance of the data here isn't always convincing. For example, the effect after 3 days of Plk1 inhibition (lines 129-130 in the text): this shows a reduction of G2/M cells from 61% to 45%. After 2 days the difference was from 69 to 57%. These drops seem relatively subtle, over such long time courses.

The decrease of the mitotic index at 72 h from 61% to 45% is a drop of more than 25%, which is statistically significant. For all three time points we analyzed statistically the difference between Δ APC-expressing and wild-type cells. We added the statistic evaluation to Figure 1 and to the text on page 5.

The scoring of cell cycle stages, from the FACS data, needs to be more carefully explained in the Figure legends.

More information about the FACS-based evaluation of cell cycle stages was added to the legend of Figure 1.

Figure 2

Why do the SW480 cells exit mitosis more slowly than the HCT116 cells, after their release from Plk1 inhibition. There is a difference of several hours, yet these are the cells that have mutated APC and the CIN phenotype. The authors need to explain and discuss these differences in timing.

Mitotic exit requires inactivation of Cyclin-dependent kinase 1 (Cdk1) and reversal of Cdk1 substrate phosphorylation. This dephosphorylation is mediated, in part by Clp1/Cdc14, a Cdk1-antagonizing phosphatase, which reverses Cdk1 phosphorylation of itself, Cdc25, and other Cdk1 substrates. PP2A^{Cdc55} prevents nucleolar release of the Cdk-antagonising phosphatase Cdc14 by counteracting phosphorylation of the nucleolar protein Net1 by Cdk. In mammalian cells, the majority of cyclin B1 must be destroyed before the cell can enter anaphase. Thus, the level of cyclin B1 is an important criterion for the timing of mitotic exit. We compared the levels of Cyclin B1 in both cell lines and observed a higher level in

SW480 cells compared to HCT116 cells, which could account for the slow exit of SW480 from mitosis.

We added a new Suppl. Fig. 3c and text to page 8, 1st paragraph.

Figure 3

It is argued that inhibition of Plk1 and Delta APC lead to endoreduplication (line 200). How do the authors know this, rather than these cells having failed cytokinesis, possibly due to defects in chromosomal passenger complex functions. Have cells been analyzed whilst undergoing cytokinesis – live-cell imaging would help here.

Our new live-cell imaging experiment showed that the expression of Δ APC in HCT116 does not induce mitotic arrest. Δ APC-expressing cells were able to complete mitosis, however, an increase in segregation failure could be observed. On the other hand, the inhibition of Plk1 led to an extensive mitotic arrest that lasted several hours followed by mitotic escape, in which the cells broke mitosis and exited with unsegregated DNA. In our experiment cells that expressed Δ APC and were treated with BI6727 showed also a long prometaphase arrest, at the end of which they underwent mitotic slippage and endoreduplication in the absence of cytokinesis. The results of the time-lapse microscopy are integrated in Figure 2 and the analysis of the data is included in page 7.

Figure 4

More explanation of these assays is needed, for the general reader. Figure legends are too brief. For example, what should we take away from the phase-contrast images (which are rather small) and the coomassie stained dishes? The ‘MTT assay’ needs a brief explanation here too.

To improve clarity, we removed the phase-contrast images, enlarged the images of the coomassie-stained dishes and counted the number of colonies. This result is summarized on page 9, 1st paragraph.

More information on the MTT assay is given in the legend of Figure 4.

Figure 6

The authors really need to look at Mps1 kinase localization and its levels at kinetochores. Surprisingly this key checkpoint kinase is rarely mentioned, and it almost certainly has key role(s) here:

- along with Plo1 it targets KNL1 for phosphorylation (and thereby Bub1/BubR1 recruitment to kinetochores)
- it also phosphorylates Bub1, leading to Mad1 recruitment to kinetochores

Mps1 kinase activity is likely co-ordinated with (and/or regulated by) Aurora B kinase activity, so Mps1 levels may well be down at these kinetochores.

This study involves direct perturbation of Plk1 activity, but Aurora B kinase and Mps1 kinase are being indirectly affected (deltaAPC) – it is not surprising that mitosis and chromosome segregation are significantly perturbed in these cell lines.

We ordered three Mps1-specific antibodies from different manufacturers, but we made the experience that all antibodies tested gave a strong background in immunofluorescence. Unfortunately, this antibody quality is not suited to answer the reviewer's questions specifically. In order to provide an answer to this question we carried out co-immunoprecipitation experiments. The details are provided in the figure and the text below.

It is already known that Plk1 and Mps1 cooperate to regulate the strength of the spindle assembly checkpoint (Von Schubert et al., Cell Reports, 2015). This paper reports that Plk1 phosphorylates at least 2 substrates of Mps1, KNL and Mps1 itself. These phosphorylations enhance Mps1 activity and are involved in the recruitment of SAC component leading to the establishment of as strong SAC. Since Mps1 phosphorylates the MELT domains of KNL1, we sought to quantify the kinetochore interaction of Mps1 with KNL1 using immunoprecipitation experiments. Therefore, we treated HCT116 and ΔAPC-expressing cells with STLC or BI6727. Then we precipitated Mps1 kinase using specific antibodies and looked for the co-precipitated amounts of KNL1. Indeed, we found that Plk1 inhibition reduced the amount of precipitated KNL1 confirming the result of other studies (see IP figure, lane 6). Interestingly, the amount of the co-precipitated KNL1 was even more reduced when cells were expressing ΔAPC and treated with BI6727 (see IP figure lane 7). The input loading

showed that the protein amounts of KNL1 were not affected, since its levels were not changed along the different conditions (see figure IP lanes 1-4). This would mean that additionally to the reduced activity of Aurora B at kinetochores of cells expressing Δ APC upon Plk1 inhibition, the recruitment and, thus the activity of Mps1 are likely altered at these sites. All these factors contribute toward weakening the SAC and are perhaps the reasons for the short mitotic arrest and the early mitotic slippage elicited by the expression of Δ APC and Plk1 inhibition.

Figure 7

Again Mps1 is a key regulator of MCC assembly, and this needs to be discussed here. MCC levels will drop as cells exit mitosis, so it is not clear here if the low levels are a cause or consequence of slippage. The authors need to block proteasome activity and repeat this experiment: are MCC levels still low?

Our immunofluorescence experiments showed that the recruitment of MCC components to kinetochore is slightly disrupted after expression of Δ APC. However, the MCC recruitment became severely impaired after adding BI6727 to Δ APC-expressing cells and both MAD1 and BuBR1 were even undetectable at certain kinetochores (Fig. 7b,c). The complementary immunoprecipitation showed a dramatic disassembly of the MCC complex upon Plk1 inhibition in Δ APC expressing cell. Still, the protein levels of the investigated MCC components in Δ APC cells treated with BI6727 were unchanged or even slightly higher compared to those of control cells in the same condition (Fig. 7d, input loading left panel). This indicates that the SAC deficiency in this case cannot be accounted for the reduction of MCC protein levels / proteasomal degradation during the arrest but it is more a result of a deficient recruitment to kinetochores and a defective assembly of the MCC. Such a deficient SAC will subsequently allow the cells to prematurely and improperly exit mitosis, hereby, generating aneuploid cells.

Figure 8

Ki67 needs better introduction here, as does generation of the plot in 8f. We can't see the Ki67 staining data clearly (Fig 8c).

We provided more information on Ki-67 in the legend of Fig. 8 and improved the quality of the figure.

Reviewer #3 (Remarks to the Author):

Within the manuscript entitled 'Plk1 has tumour-suppressive potential in APC-truncated colon cancer cells' the authors major claim is that Plk1 along with APC play an important role in suppressing tumourigenesis and upon mutation of APC, inhibition of Plk1 results in silencing of the SAC so promoting a premature exit from the cell cycle resulting in an increase in aneuploidy. The cell culture and in vivo models used in this manuscript show that inhibiting or knocking down Plk1 in conjunction with truncated APC results in an increase in cell division, polyploidy and small intestinal tumours which correlate with human data where CRC patients with low Plk1 have a poor survival. This suggests that any inhibitors of Plk1 would not have a clinical benefit in a CRC cohort. The role of Plk1 in tumourigenesis has previously been discussed where it was shown that Plk1-/+ mice developed spontaneous tumours more rapidly than wild-type counterparts. However, no previous work has focussed on the role of Plk1 following the loss of APC. Expression of truncated APC in colon cancer cells overrides partially the mitotic arrest induced by Plk1 inhibition. Overall there are some interesting points to the manuscript however there are many issues with the paper (detailed below, not least with data presentation and analysis) which preclude publication.

Specific comments

1. Throughout the paper, the authors use the small molecule BI6727 as an inhibitor of Plk1, however, they show in figure 1 that this protein is still present (and increases in expression following treatment with BI6727). They should demonstrate the effect of this inhibitor on the activity of Plk1/its downstream effector (or referencing how it works).

To address this issues, we have performed two types of experiments:

1. *in vitro* kinase assays (a): We added increasing amounts of the ATP-competitive inhibitor BI6727 *in vitro* to the Plk1 protein and monitored the autophosphorylation of Plk1. Even at a low concentration of 0.1 nM the autophosphorylation is clearly reduced indicating the high potency of this inhibitor, which is used for many clinical trials.

2. *in vivo* kinase assays (b): In this case, the situation is more complex. A population treated with this inhibitor (even at low concentrations) shows an enrichment of cells in G2/M, which express Plk1 at high levels. Therefore, when we precipitated Plk1 with Plk1-specific antibodies (WB), we see at low concentrations an increase in the amount of Plk1 in lysates of inhibitor-treated cells. At higher concentrations the level of Plk1 decreases due to cell death. If we use immunoprecipitated Plk1 from inhibitor-treated cells for an *in vivo* kinase assay, the

inhibition of Plk1, as indicated by the Histone H1 phosphorylation, becomes apparent at inhibitor concentrations >50 nM (see figure below).

2. Members of the Plk family are well known regulators of cell cycle progression including mitosis and it is established that PLK function is associated with cancer1. However, it is not clear if activity of Plk1 is altered upon loss of function of APC. From figure 1 it appears that expression of Plk1 is unaltered following expression of truncated APC. Does the activity of Plk1 change following expression of truncated APC. How is APC controlling Plk1 activity? Is it WNT dependent? The authors have not shown a direct link between Plk1 and APC – at present this link is assumed.

We treated Δ APC-expressing cells and wild-type cells with Nocodazole for an enrichment of the cell population in G2/M and to obtain strong Plk1 expression. In those cells we monitored Myt1, which is phosphorylated by Plk1 at Thr-495. Expression of Δ APC did not change the levels of Myt1 phosphorylation (pT495) in mitotic cells compared to mitotic wild-type cells suggesting that Δ APC expression does not alter the activity of Plk1 in cells (see figure below).

3. The exit of cells from a mitotic arrest induced by Plk1 inhibition is accelerated by truncated APC expression

The finding that loss of function of APC has no effect on proliferation of either cell line (figure S2) is perhaps surprising given that it is well established that loss of APC results in an increased proliferation in vivo². It would be interesting to see analysis of downstream WNT targets to determine how the truncated APC transgene is affecting well known pathways.

We added new data to Suppl. Fig. 2a (right panel) representing the long-term examination of cells with loss of APC function. The extended analysis with an observation period of 12 days confirmed the increased proliferative activity of cells expressing truncated APC.

The results of the examination of downstream WNT targets are shown in Fig. 4d.

4. Impact of truncated APC expression on polyploidy and survival of Plk1 inhibitor-treated colon cancer cells

Throughout the manuscript the quality of images is very poor making it difficult to interpret some data.

We are sorry for the poor quality of the images. All IF images were generated using a confocal laser scanning microscope and should have a very high resolution. The conversion to pdf has altered the resolution of the images. For the new version of the manuscript we have improved the resolution of the figures.

The text surrounding figure 4 talks about crystal violet staining, where the figure legend mentions coomassie staining – this should be consistent. The authors state that more floating cells and debris was observed in the truncated-APC cultures however, no data was shown to demonstrate this (this may be present in the figures, however due to the quality –I cannot see this).

Indeed, colonies were stained with Coomassie Blue. The corresponding text was corrected (page 9, top). To improve the clarity of the figure, we removed the phase-contrast images and enlarged the images representing Coomassie stained-dishes.

It is stated in the text that more caspase and annexin was observed in the controls vs APC counterparts at a range of concentrations, however the data show that at 24 hours more caspase was observed in the truncated APC cell lines.

At an early time-point of BI6727 incubation (24 h) there is more caspase 3/7 activity in Δ APC-expressing cells. This difference seems to be relatively subtle. At least in SW480 cells this difference doesn't translate to more cell death based on the Annexin staining

(Supplementary Fig. 4b, lower panel). In long-term experiments the anti-apoptotic impact of Δ APC expression is convincing regarding for example Figure 4 or the animal experiments (Fig. 8)

From the proliferation curves, it is clear that inhibition of Plk1 in cell expressing truncated APC results in an increase in cell proliferation.

5. The inhibition of Plk1 in truncated APC-expression cells increases chromosome abnormalities

Figure 5 again has poor quality images of the metaphase spreads (with no scale bar on fig S4a).

A scale bar was introduced into Suppl. Fig. 5a and mentioned in the corresponding legends of Suppl. Fig. 5a and Fig. 5.

In addition to this, the axis on the chromosome number histograms are all inconsistent making it difficult to compare and interpret.

The data was generated with GraphPad Prism software and the authors have no influence on how the data is generated and displayed.

However, it is clear from the analysis shown in figure 5c that chromosome number/cell increases when Plk1 is inhibited in cells expressing truncated APC.

6. Inhibiting Plk1 in truncated APC expressing cells reduces the protein level and activity of Aurora B at Kinetochores

In figures 6, 7, S5 and S6 – there is no IF carried out on non-treated cells. How is the localisation of Aurora B, H3 pT3, CENP-A P27 and BubR1 affected by BI6727 and loss of function of APC? Whilst it has been shown that inhibiting Plk1 and/or APC function reduces that amount of AuroraB, H3pT3, CENPA pS7, BubR1 and Mad1m, there is no data to show how these are altered from a control, untreated sample.

In all treatment groups depicted in Fig. 6, 7 and Suppl. Fig. 6, 7 cells were incubated with BI6727 at a concentration of 100 nM, which increases the percentage of cells in G2/M to approximately 75%. Our control cells were treated with a reversible Eg5 inhibitor, STLC that enriches also cells in G2/M. STLC induces a prometaphase arrest without interfering with microtubule dynamics. Since both inhibitors, BI6727 and STLC, induce a prometaphase arrest, we induce a comparable situation for the quantitative examination of Aurora B, H3pT3, CENPA pS7, BubR1 and Mad1. In untreated cell populations we would hardly be

able to identify a high number of mitotic cells that are needed for the statistical analysis of our findings.

Additionally – all of the plots have been scatter dot plots except the Figure 6c – is there a reason for this?

There is no particular reason. We just preferred to use graphical representation of the results in this case

**7. The inhibition of Plk1 increases the frequency of colonic tumours in APCMin/+ mice
The authors say that inhibition of Plk1 increases the frequency of colonic tumours in APCmin mice. However, they do not show that in this manuscript. They do however, show an increase in tumour in the small intestine.**

Thank you very much for this important aspect. To address this point, we have reworded the headline for this section and the corresponding sentences on page 12 and within the discussion.

Additionally, as with previous figures, the quality of the images is not sufficient to make any claims regarding the pathology of tumour.

For the tumor sections we have improved the resolution of the figures.

The authors claim that the tumours in the APCMin mice was invasive into the mucosa, however, this is not clear from the pictures provided.

We have also introduced arrows to highlight the invasion into the mucosa.

The legend for Fig. 8c was extended.

There are several claims regarding proliferation and Ki67 staining, however, from the images provided, I cannot see any Ki67 staining.

Each section stained with hematoxylin is accompanied with an image representing the KI67 staining in Fig. 8. The score of the KI staining of intestinal tumor sections is shown Fig. 8f.

We are presenting improved figures

Authors have used commas as opposed to decimal points on graphs.

Commas were replaced by decimal points.

The authors do show that the number of small intestinal tumours are significantly increased following inhibition of Plk1 in an APC^{min/+} model of CRC, which hasn't been shown previously.

8. Patients with colon tumours expressing low Plk1 have a worse clinical outcome
The authors show that in a human cohort of CRC patients, that high levels of Plk1 is significantly tumour suppressive (however the P value on the graph is incorrect). This is in contrast to a previous study where it has been shown that PLk1 is overexpressed in CRC cancer specimens and is linked to progression of a primary CRC tumour³.

As the reviewer indicates, other groups have found a positive correlation of PLK1 to tumour progression. It is important to note that these studies examined all colon tumours irrespective of APC mutation status.

Our study specifically examines PLK1 expression (high versus low) in a patient cohort of APC mutant colorectal cancers. Thus, our data is not conflicting with the literature but highlight the epistatic nature of PLK1 function in colon cancer.

p-values for our Kaplan Meier curves were generated using TCGA survival data in Prism 6. We have used log-rank (Mantel-Cox) test to determine significance (p-value). We follow standard statistical methodology and add the following reference to our manuscript describing the statistical use of the log-rank test for survival analysis (Bewick et al., Statistics review 12: Survival analysis, *Crit Care*. 2004; 8(5): 389–394). This reference was added to the Bioinformatics analysis methods on page 14. In addition, we included the prism data files containing the survival analysis as new Suppl. Tab. 1.

9. Throughout the manuscript all statistical tests were carried out using students T-test, unpaired and two-tailed, which is not the best statistical test for some of the experiments (for instance in figures 3, 5, 6, 7) Mann-whintey would be more appropriate. Additionally, no exact P values were provided. No catalogue or clone numbers for antibodies are provided, although suppliers are mentioned.

Mann-Whitney *U* Test for statistics has been carried out for Figures 2,5,6,7 and Supp. Figures 5,6,7,9.

The catalogue clone numbers for the antibodies were introduced into material and methods part.

Overall, I think the authors have shown that inhibition of Plk1 in exacerbates the loss of function of APC phenotype particularly in the context of cell proliferation, aneuploidy and tumour formation, which has not been shown before. My concerns regarding this manuscript are that the figures are of very poor quality making it difficult to independently assess the data and no direct link between APC and Plk1 has been shown.

Reviewers' comments:

Reviewer #1 (Remarks to the Author):

The authors have adequately addressed all of my concerns. The revised version is much improved. I support the publication of this manuscript.

Reviewer #2 (Remarks to the Author):

The authors have done a good job at addressing most of my criticisms, and I am now happy to recommend publication, after minor changes.

Figure 1: I still think that the Δ APC terminology might confuse a general reader (as evidenced by multiple reviewers' confusion on this point!). The text on page 5 is now much better. My further suggestion is two fold:

1) rather than simply calling this Δ APC (which to many geneticists reads like a complete deletion, or null mutant) why not use APC- Δ Cterm or APC- Δ C ? I think that this would more clearly describe APC with its C-terminus deleted.

2) Also/alternatively: add a simple schematic diagram to Fig 1A (next to the western blot). This should indicate which functional domains of APC are present and which are not (in this C-terminal deletion/truncation).

Figure 2: This is much improved - the live cell imaging is a significant addition to this manuscript, and clearly describes the mitotic behaviours of the different cell lines.

In general the figure legends, statistics and description of various assays has been significantly improved, as requested by reviewers.

Figure 6: I thank the authors for carrying out these Mps1 experiments. It is a shame that the immunofluorescence experiments were not possible, due to the available reagents. Perhaps they could mention the reduced association, between KNL1 and Mps1, in the discussion?

Figure 7: I think the authors mis-understood me here. I was not arguing that the MCC components (individual proteins) would be degraded upon mitotic exit/slippage. Rather that the MCC complex would be dis-assembled once CDK activity dropped. It is very hard to know whether reduced levels of the MCC complex is a cause or consequence of slippage. I am happy to accept that the SAC is generally 'defective' here, presumably due to significantly perturbed kinetochore structures.

Reviewer #3 (Remarks to the Author):

I would like to thank the authors for making the amendments requested and for improving the quality of the images.

Overall, I think whilst the paper contains some interesting data I still have some concerns regarding this manuscript.

The authors have shown, using truncated APC as a driver of colon cancer that this, alongside loss of Plk1 results in more aneuploidy and more tumours. However, it is unclear if the authors believe that Plk1 and APC are directly mechanistically linked (in which case, this link must be shown), or if this is just an additive effect. Apc loss is known to have effects on mitosis so does PLK1 loss have more impact on APC deficient than wild type cells?.

My major concern with this paper remains the in vivo work particularly the histological/pathological characterisation of the adenoma e.g. higher and lower grade adenomas. Examples of both should be provided so the readers know what is being referred to. Additionally, from the images provided, there are no examples of a 'higher grade' adenoma. Most of the examples are polyps. Where the arrow is pointing to in 8c (right panel, ApcMin/+ with and without BI6726), this is not a feature of an advanced adenoma, this is cross cut through normal intestinal crypts. A pathologist needs to be involved in the characterisation of the lesion and cross cut lesions should be avoided.

Other points:

- The annotation is incorrect throughout the images and paper a capital letter and italics should be used for mouse genes.
 - In Figure 8, the authors refer to black arrows several times and across several images. It would be useful if the distinction as to which arrow corresponds to which statement (either change colour or use a different symbol).
 - The Ki67 staining is not very clear. This could be due to the strong counterstain – the counter stain looks more like a H&E rather than haematoxylin alone.
- . In figure 5, I think it would be beneficial to the reader if the axis were changed to be consistent. This can be achieved in GraphPad Prism by right clicking on the axis once the graph has been generated, and selecting format axis.
- . Additionally, I think it would provide more transparency if figure 6c were made into a scatter plot, especially as all the other plots of this type are scatter plots. The use of bar charts can sometimes be used to hide outliers and clusters.

It would be useful to have the exact P value rather than >0.05.

We thank the reviewers for the valuable feedback in response to our manuscript entitled "Plk1 has tumor-suppressive potential in APC-truncated colon cancer cells". In response to the concerns raised by the reviewers, we made certain insertions and corrections to the manuscript, regarding both, the experimental work and the written material. Changes in the manuscript are underlined.

Reviewers' comments:

Reviewer #1 (Remarks to the Author):

The authors have adequately addressed all of my concerns. The revised version is much improved. I support the publication of this manuscript.

Reviewer #2 (Remarks to the Author):

The authors have done a good job at addressing most of my criticisms, and I am now happy to recommend publication, after minor changes.

Figure 1: I still think that the Δ APC terminology might confuse a general reader (as evidenced by multiple reviewers' confusion on this point!). The text on page 5 is now much better. My further suggestion is two fold:

1) rather than simply calling this Δ APC (which to many geneticists reads like a complete deletion, or null mutant) why not use APC- Δ Cterm or APC- Δ C ? I think that this would more clearly describe APC with its C-terminus deleted.

Throughout the manuscript we changed Δ APC to APC- Δ C.

2) Also/alternatively: add a simple schematic diagram to Fig 1A (next to the western blot). This should indicate which functional domains of APC are present and which are not (in this C-terminal deletion/truncation).

We added a schematic diagram to Fig. 1a.

Figure 2: This is much improved - the live cell imaging is a significant addition to this manuscript, and clearly describes the mitotic behaviours of the different cell lines.

In general the figure legends, statistics and description of various assays has been significantly improved, as requested by reviewers.

Figure 6: I thank the authors for carrying out these Mps1 experiments. It is a shame that the immunofluorescence experiments were not possible, due to the available reagents.

Perhaps they could mention the reduced association, between KNL1 and Mps1, in the discussion?

We added a short paragraph to the Discussion on page 17 to address this aspect.

Figure 7: I think the authors mis-understood me here. I was not arguing that the MCC components (individual proteins) would be degraded upon mitotic exit/slippage. Rather that the MCC complex would be dis-assembled once CDK activity dropped. It is very hard to know whether reduced levels of the MCC complex is a cause or consequence of slippage. I am happy to accept that the SAC is generally 'defective' here, presumably due to significantly perturbed kinetochore structures.

Reviewer #3 (Remarks to the Author):

I would like to thank the authors for making the amendments requested and for improving the quality of the images.

Overall, I think whilst the paper contains some interesting data I still have some concerns regarding this manuscript.

The authors have shown, using truncated APC as a driver of colon cancer that this, alongside loss of Plk1 results in more aneuploidy and more tumours. However, it is unclear if the authors believe that Plk1 and APC are directly mechanistically linked (in which case, this link must be shown), or if this is just an additive effect. Apc loss is known to have effects on mitosis so does PLK1 loss have more impact on APC deficient than wild type cells?.

We discussed this aspect in detail on pages 16 and 17.

My major concern with this paper remains the in vivo work particularly the histological/pathological characterisation of the adenoma e.g. higher and lower grade adenomas. Examples of both should be provided so the readers know what is being referred to. Additionally, from the images provided, there are no examples of a 'higher grade' adenoma.

We labelled low and high-grade adenomas.

Most of the examples are polyps. Where the arrow is pointing to in 8c (right panel, ApcMin/+ with and without BI6726), this is not a feature of an advanced adenoma, this is cross cut through normal intestinal crypts.

We improved the labeling to make sure that arrows correlate with the corresponding text.

A pathologist needs to be involved in the characterisation of the lesion and cross cut lesions should be avoided.

We discussed Fig. 8 in detail with our pathologist Dr. Hörlin and made several changes.

Other points:

- **The annotation is incorrect throughout the images and paper a capital letter and italics should be used for mouse genes.**

Throughout the text all annotations were changed.

- **In Figure 8, the authors refer to black arrows several times and across several images. It would be useful if the distinction as to which arrow corresponds to which statement (either change colour or use a different symbol).**

We discussed Fig. 8 in detail with our pathologist Dr. Hörlin and made the following changes. To distinguish between individual images belonging to Fig. 8c, we labelled the images from A-F. The description of the histological images on page 14 (1. paragraph) addresses Fig. 8c, A-F in detail. Now it should be absolutely clear which arrow belongs to statement.

- **The Ki67 staining is not very clear. This could be due to the strong counterstain – the counter stain looks more like a H&E rather than haematoxylin alone.**

For Ki67 staining we used a dye-tagged antibody. The sections were colored with Fast Red (Permanent AP Red Kit, Zytomed) and counterstained with hematoxylin. Fast growing tumors were stained intensively with the red dye followed by the blue counterstain (hematoxylin). The resulting images show a purple color.

- . **In figure 5, I think it would be beneficial to the reader if the axis were changed to be consistent. This can be achieved in GraphPad Prism by right clicking on the axis once the graph has been generated, and selecting format axis.**

Fig. 5, We changed the y-axis to be consistent.

- . **Additionally, I think it would provide more transparency if figure 6c were made into a scatter plot, especially as all the other plots of this type are scatter plots. The use of bar charts can sometimes be used to hide outliers and clusters.**

Fig. 6 c was transformed into a scatter plot.

REVIEWERS' COMMENTS:

Reviewer #3 (Remarks to the Author):

The authors have provided further clarity and answered my comments either by additions to the text or by improving the quality of their figures. I'm pleased that they have now shown their images to a pathologist. I still think some of the panels in figure 8 e.g. show sections that are cross cut and a better quantification of proliferation e.g. BrdU would have been more helpful. However given the many improvements during the course of the review processes I believe this manuscript is now acceptable.

Minor comment

The authors should mention what the arrows represent in the figure legend.

We thank the reviewer for the helpful feedback in response to our manuscript entitled “Plk1 has tumor-suppressive potential in APC-truncated colon cancer cells”. In response to the concerns raised by the reviewer, we made an insertion to the manuscript. Changes in the manuscript are underlined.

Reviewers' comments:

Reviewer #3 (Remarks to the Author):

The authors have provided further clarity and answered my comments either by additions to the text or by improving the quality of their figures. Im pleased that they have now shown their images to a pathologist. I still think some of panels in figure 8 e.g show sections that are cross cut and a better quantification of proliferation e.g. Brdu would have been more helpful. However given the many improvements during the course of the review processes i believe this manuscript is now acceptable.

Minor comment

The authors should mention what the arrows represent in the figure legend.

We added the corresponding information to the legend.